# Perturbation-based Regret Analysis of Predictive Control in Linear Time Varying Systems

**Yiheng Lin**
California Institute of Technology
Pasadena, CA, USA
yihengl@caltech.edu

**Yang Hu**
Tsinghua University
Beijing, China
huy18@mails.tsinghua.edu.cn

**Guanya Shi**
California Institute of Technology
Pasadena, CA, USA
gshi@caltech.edu

**Haoyuan Sun**
California Institute of Technology
Pasadena, CA, USA
hsun2@caltech.edu

**Guannan Qu**
California Institute of Technology
Pasadena, CA, USA
gqu@caltech.edu

**Adam Wierman**
California Institute of Technology
Pasadena, CA, USA
adamw@caltech.edu

## Abstract

We study predictive control in a setting where the dynamics are time-varying and linear, and the costs are time-varying and well-conditioned. At each time step, the controller receives the exact predictions of costs, dynamics, and disturbances for the future $k$ time steps. We show that when the prediction window $k$ is sufficiently large, predictive control is input-to-state stable and achieves a dynamic regret of $O(\lambda^k T)$, where $\lambda < 1$ is a positive constant. This is the first dynamic regret bound on the predictive control of linear time-varying systems. We also show a variation of predictive control obtains the first competitive bound for the control of linear time-varying systems: $1 + O(\lambda^k)$. Our results are derived using a novel proof framework based on a perturbation bound that characterizes how a small change to the system parameters impacts the optimal trajectory.

## 1 Introduction

We study the problem of predictive control in a linear time-varying (LTV) system, where the dynamics is given by $x_{t+1} = A_t x_t + B_t u_t + w_t$. Here, $x_t$ is the state, $u_t$ is the control action, and $w_t$ is the disturbance or exogenous input. At each time step $t$, the online controller incurs a time-varying state cost $f_t(x_t)$ and control cost $c_t(u_{t-1})$, and then decides its next control action $u_t$. In deciding $u_t$ the controller makes use of predictions of the next $k$ future disturbances, cost functions, and dynamical matrices, and seeks to minimize its total cost on a finite horizon $T$. Our main results bound the dynamic regret and competitive ratio of predictive controllers in this LTV setting.

Recently, a growing literature has sought to design controllers that achieve learning guarantees such as static regret [1, 2], dynamic regret [3, 4], and competitive ratio [5]. The most relevant

---

[†]Yiheng Lin, Yang Hu, Haoyuan Sun, Guanya Shi, and Guannan Qu contributed equally to this work.

[‡]This work was supported by NSF grants CNS-2106403, NGSDI-2105648, and AitF-1637598, with additional support from Amazon AWS, PIMCO, and the Resnick Sustainability Insitute. Yiheng Lin was supported by Kortschak Scholars program.

line of work concerns predictive control with learning guarantees, which studies how to leverage the prediction window $k$ to reduce the regret and competitive ratio. This line of work has focused on linear time-invariant (LTI) systems [3, 4, 6, 7]. However, linear time-varying (LTV) systems have received increasing attention in recent years due to their importance in a variety of emerging applications, despite the challenges associated with analysis. For example, in the problem of power grid frequency regulation, the dynamics is determined by the proportion of renewable energy in total power generation, which is time-varying [8, 9]. It is also common to use the LTV systems as an approximation of nonlinear dynamics in predictive control and planning [10–13].

The current lack of progress toward understanding measures like regret and competitive ratio in LTV settings is due to the need for new techniques to generalize the dynamics from LTI to LTV and the costs from quadratic to well-conditioned functions. Specifically, the proof approaches used in previous studies on regret and competitive ratio of predictive control in LTI dynamics with quadratic costs, e.g., [4, 6, 7], require explicitly writing down the cost-to-go function, optimal control actions, and algorithm's actions as functions of the system parameters. This is very difficult, if not impossible, for general cost functions that do not have a quadratic form. A promising approach that does not require such explicit characterizations is to derive results via reductions from optimal control to online convex optimization with multi-step memory, e.g., [1, 3, 5, 14, 15]. However, such reductions usually do not work well for LTV systems due to the need to represent the problem in control canonical form [3, 5], or due to limitations on the policy class and comparisons to static benchmarks [1, 15].

Perhaps the most prominent approach for controlling LTV systems is Model Predictive Control (MPC), also known as Receding Horizon Control [16]. Generally speaking, at each time step, an MPC-style algorithm solves a predictive trajectory for the future $k$ time steps and commit the first control action in this trajectory. MPC-style algorithms are known to work well in practice, even when the dynamics are non-linear and time-varying, e.g., [13, 17–19]. On theoretical side, the asymptotic behaviors of MPC such as stability and convergence have been studied intensively under general assumptions on dynamics and costs [20–23]. However, non-asymptotic guarantees such as regret and competitive ratio of MPC-style policies have been limited. Despite recent work providing such guarantees in the context of LTI systems with quadratic costs, e.g., [4, 6, 7], the derivation of regret and competitive ratio results for MPC in LTV systems remains open.

**Contributions.** We provide the first regret and competitive ratio results for a controller in LTV systems with time-varying costs. Specifically, we show that an MPC-style predictive control algorithm (Algorithm 1) achieves a dynamic regret that decays exponentially with respect to the length of prediction window $k$ in the LTV system (Theorem 4.2): $O(\lambda^k T)$, where the decay rate $\lambda$ is a positive constant less than 1. This almost matches the exponential lower bound for improvement from predictions in the LTI setting shown in [3] in the sense that, to achieve any target regret level, the required length of prediction $k$ shown by our bound differs from the theoretical lower bound by at most a constant factor. With a variation of predictive control (Algorithm 2), we also show the first competitive bound in LTV systems with time-varying well-conditioned costs (Theorem 4.3): $1 + O(\lambda^k)$, where the decay rate $\lambda$ is identical with the one in the regret bound.

We develop a novel analysis framework based on a perturbation approach. Specifically, instead of solving for the optimal states and control actions like previous analyses in the LTI setting with quadratic costs, e.g., [4, 7], we bound how much impact an perturbation to the system parameters can have on the optimal solution. This type of perturbation bound (Theorem 3.3) can be shown even when the optimal trajectory cannot be written down explicitly, which allows it to be applied in LTV systems with well-conditioned costs. Then, we utilize this perturbation bound to establish results on dynamic regret and the competitive ratio. In addition, we want to emphasize that the perturbation approach we develop is highly modular and extendable. For instance, if a stronger perturbation bound for some specific class of dynamics and/or cost functions can be shown, the dynamic regret of the predictive controller will improve. Similarly, to further generalize the problem setting (e.g., to include additional constraints), one only needs to establish the corresponding perturbation bounds and the regret result will follow.

Another important component of the proof is a novel reduction between LTV control and online optimization. Connections between online optimization and control have received increasing attention in recent years, e.g., [1, 3, 5, 14, 15]. Existing reductions rely on the canonical form, which does not apply to LTV systems, and/or formulations of online optimization with memory of multiple prior time steps, which makes the online problem more challenging. The reduction we present here relies

on neither, and is thus a fundamentally different approach to connect control and online optimization. Further, this reduction is not specific to the predictive control algorithm we study, and we expect it to prove useful for other controllers in future work. A limitation of our reduction framework is that it cannot handle state/control constraints. This limitation is shared by previous works [3, 4, 6, 7], and represents a challenging open question in the literature.

## 2 Background and Setting

We consider a finite-horizon discrete-time online control problem with linear time-varying (LTV) dynamics, time-varying costs, and disturbances, namely

$$\min_{x_{0:T}, u_{0:T-1}} \sum_{t=1}^{T} (f_t(x_t) + c_t(u_{t-1}))$$
$$\text{s.t. } x_t = A_{t-1}x_{t-1} + B_{t-1}u_{t-1} + w_{t-1}, t = 1, \ldots, T, \tag{1}$$
$$x_0 = x(0),$$

where $x_t \in \mathbb{R}^n$, $u_t \in \mathbb{R}^m$, and $w_t \in \mathbb{R}^n$ respectively denote the state, the control action, and the disturbance of the system at time steps $t = 1, \ldots, T$, and $x(0) \in \mathbb{R}^n$ is a given initial state. By convention, the hitting cost function $f_t : \mathbb{R}^n \to \mathbb{R}_+$ and control cost function $c_t : \mathbb{R}^m \to \mathbb{R}_+$ are assumed to be time-varying and well-conditioned. Define the tuple $\vartheta_t := (A_t, B_t, w_t, f_{t+1}, c_{t+1})$.

In the classical setting where no predictions are available, after observing state $x_t$ at time step $t$, the algorithm needs to decide the control action $u_t$ before observing $\vartheta_t$, which is an unknown random disturbance input. We use the following event sequence to describe this ordering:

$$x_0, u_0, \vartheta_0, x_1, u_1, \vartheta_1, x_2, \ldots, x_{T-1}, u_{T-1}, \vartheta_{T-1}, x_T.$$

We assume that the algorithm has access to the exact predictions of disturbances, cost functions and dynamical matrices in the future $k$ time steps (which are time-varying); i.e., the event sequence is

$$x_0, \vartheta_0, \vartheta_1, \ldots, \vartheta_{k-1}, u_0, \vartheta_k, u_1, \vartheta_{k+1}, \ldots, u_{T-k-1}, \vartheta_{T-1}, u_{T-k}, u_{T-k+1}, \ldots, u_{T-1}.$$

Here we assume all predictions are *exact*, and leave the case of inexact predictions for future work. This prediction model has been used in previous works like [4, 24–26], and is available in many real-world applications such as disturbance estimation in robotics and frequency regulation in power grids. The availability is due to the fact that, in such scenarios as mentioned above, experiments or observations on the dynamics can be conducted repeatedly and consistently, which makes it feasible to train a good predictor based on the data collected from repeated trials.

### 2.1 Assumptions

As is standard in studies of regret and competitive ratio in linear control problems, we assume the cost functions are well-conditioned.

**Assumption 2.1** (Well-conditioned Costs). *The cost functions satisfy the following constraints:*

1. *$f_t(\cdot)$ is $m_f$-strongly convex for $t = 1, \ldots, T$, and $\ell_f$-strongly smooth for $t = 1, \ldots, T - 1$.*

2. *$c_t(\cdot)$ is both $m_c$-strongly convex and $\ell_c$-strongly smooth for $t = 1, \ldots, T$.*

3. *$f_t(\cdot)$ and $c_t(\cdot)$ are twice continuously differentiable for $t = 1, \ldots, T$.*

4. *$f_t(\cdot)$ and $c_t(\cdot)$ are non-negative, and $f_t(0) = c_t(0) = 0$ for $t = 1, \ldots, T$.*

Note that assumptions (1) through (3) are quite common [3, 5, 14, 25, 27]. Assumption (4) is less common, but can be satisfied via re-parameterization without loss of generality. Specifically, when the minimizers of state cost $f_t$ and control cost $c_t$ are nonzero, we perform the transformation

$$x'_t \leftarrow x_t - \arg\min_x f_t(x), \ u'_t \leftarrow u_t - \arg\min_u c_{t+1}(u),$$
$$w'_t \leftarrow w_t + A_t \arg\min_x f_t(x) + B_t \arg\min_u c_{t+1}(u).$$

The intuition of this transformation is that, when the minimizer of the cost function for the next step is known, we can always perform a translation in the state and control space to align the minimizer

with the origin. We refer the interested readers to Example 2.1 for a more intuitive explanation of the above transformation.

Additionally, we need to assume the dynamics are *controllable*. It is crucial that the dynamical system can be steered from an arbitrary initial state to an arbitrary final state via a finite sequence of admissible control actions. For linear time-invariant (LTI) systems, the full-rankness of the *controllability matrix* completely characterizes the reachability of the state space, which is generally used as a standard assumption for analysis [7, 28, 29]. This can be generalized to parallel assumptions for LTV systems as follows. We begin with a definition.

**Definition 2.1.** *For a dynamical system with linear time-varying dynamics $x_t = A_{t-1}x_{t-1} + B_{t-1}u_{t-1} + w_{t-1}$, $t = 1, \ldots, T$, the transition matrix $\Phi(t_2, t_1) \in \mathbb{R}^{n \times n}$ (from time step $t_1$ to $t_2$) is defined as*

$$\Phi(t_2, t_1) := \begin{cases} A_{t_2-1}A_{t_2-2}\cdots A_{t_1} & \text{if } t_2 > t_1 \\ I & \text{if } t_2 \leq t_1 \end{cases},$$

*and the controllability matrix $M(t, p) \in \mathbb{R}^{n \times (mp)}$ is defined as*

$$M(t, p) := \big[\Phi(t + p, t + 1)B_t, \Phi(t + p, t + 2)B_{t+1}, \ldots, \Phi(t + p, t + p)B_{t+p}\big].$$

*The dynamical system is called controllable if there exists a constant $d \in \mathbb{Z}_+$, such that the controllability matrix $M(t, d)$ is of full row rank for any $t = 1, \ldots, T - d$. The smallest constant $d$ with such property is called the controllability index of the system.*

Given the above definition, we can state the key assumption necessary for the analysis of LTV systems. We use a slightly stronger assumption than being merely controllable, which we refer to as $(d, \sigma)$-uniform controllability. It is a natural generalization of its counterpart for LTI systems (see Assumption 2 in [28], where $(d, \sigma)$ is instead named as $(\ell, \nu)$).

**Assumption 2.2.** *There exists positive constants a, b, and b′, such that*

$$\|A_t\| \leq a, \ \|B_t\| \leq b, \ \text{and} \ \|B_t^{\dagger}\| \leq b'$$

*hold for all time steps $t = 0, \ldots, T - 1$, where $B_t^{\dagger}$ denotes the Moore–Penrose inverse of matrix $B_t$. Furthermore, there exists a positive constant $\sigma$ such that*

$$\sigma_{\min}(M(t, d)) \geq \sigma$$

*holds for all time steps $t = 0, \ldots, T - d$, where d denotes the controllability index.*

Note that Assumption 2.2 implies $\sigma_{\min}(M(t, p)) \geq \sigma$ for all $p \geq d$ because appending more columns to a matrix with full row rank will not reduce its minimum singular value.

The LTV setting we consider is more general than the settings which existing results on regret and competitive ratio have assumed [1, 3, 4, 7]. We highlight the implications of this general setting for enabling applications in the following examples.

**Example 2.1** (Trajectory tracking in LTV systems with well-conditioned costs)**.** *Consider a trajectory tracking problem with LTV dynamics and well-conditioned costs, which generalizes the standard linear quadratic tracking problem in [4, 30] with LTI dynamics and quadratic costs. We adopt LTV dynamics $x_{t+1} = A_t x_t + B_t u_t + w_t$ and general well-conditioned cost functions $f_t(\cdot), c_t(\cdot)$ (see Assumption 2.1). With the desired trajectory $d_{1:T}$, we consider a new state $\tilde{x}_t := x_t - d_t$ and a new disturbance $\tilde{w}_t := w_t + A_t d_t - d_{t+1}$. Thus, using the new state and disturbance, the problem naturally fits into our problem setting with k future predictions of $(A_t, B_t, w_t, f_t, c_t, d_{t+1})$. Note that predictive control with LTV dynamics is practical in nonlinear systems [13, 31] because the nonlinearity could be well approximated by LTV models [31].*

**Example 2.2** (Power grid frequency regulation)**.** *Consider the frequency regulation problem in [8], where state $x = [\theta^{\top}, \omega^{\top}]^{\top}$ represent the status of a power plant, and power generation $p_{in} \in \mathbb{R}^n$ is the control action. The continuous-time dynamics is given by*

$$\underbrace{\begin{bmatrix} \dot{\theta} \\ \dot{\omega} \end{bmatrix}}_{\dot{x}(t)} = \underbrace{\begin{bmatrix} 0 & I \\ -M(t)^{-1}L & -M(t)^{-1}D \end{bmatrix}}_{\hat{A}(t)} \underbrace{\begin{bmatrix} \theta \\ \omega \end{bmatrix}}_{x(t)} + \underbrace{\begin{bmatrix} 0 \\ M(t)^{-1} \end{bmatrix}}_{\hat{B}(t)} \underbrace{p_{in}}_{u(t)}.$$

---

**Algorithm 1** Predictive Control ($PC_k$)

---

1: **for** $t = 0, 1, \ldots, T - k - 1$ **do**
2:     Observe current state $x_t$ and receive predictions $\vartheta_{t:t+k-1}$.
3:     Solve and commit control actions $u_t := \tilde{\psi}_t^k(x_t, w_{t:t+k-1}; F)_{v_0}$.
4: At time step $t = T - k$, observe current state $x_t$ and receive predictions $\vartheta_{t:T-1}$.
5: Solve and commit control actions $u_{t:T-1} := \tilde{\psi}_t^k(x_t, w_{t:T-1}; 0)_{v_{0:k-1}}$.

---

*Here $M(t)$ denotes the rotational inertia matrix, which is time-varying and is determined by the proportion of renewable power in total power generation at time $t$, and can be accurately predicted in a certain time horizon [32, 33]; L and D are known system parameters. Using standard discretization techniques, we can formulate a discrete-time linear time-varying system $x_{t+1} = A_t x_t + B_t u_t + w_t$, where $A_t$ and $B_t$ are determined by $\hat{A}(t)$ and $\hat{B}(t)$. The cost functions are quadratic costs which penalizes frequency deviation [8]. This setting fits into our predictive control algorithm, since the controllers have accurate predictions of $A_t$ and $B_t$ in the near future due to predictablity of $M(t)$.*

## 2.2 Predictive Control

We study a classical predictive control (PC) algorithm inspired by model predictive control, which solves the optimization problem of $k$ future time steps (where $k$ is called the *prediction window*). specifically, the algorithm receives the dynamics and disturbances of the next $k$ time steps, calculates the optimal solution, and then applies the first control action of the optimal solution. The PC algorithm with prediction window $k$ is denoted as $PC_k$.

More formally, At time step $t < T - k$, $PC_k$ solves the optimization problem $\tilde{\psi}_t^k(x_t, w_{t:t+k-1}; F)$. Since we need to consider horizon lengths other than $k$, for arbitrary $p \geq 1$ and time step $t$, we define the optimization problem $\tilde{\psi}_t^p(x, \zeta; F)$ as

$$\tilde{\psi}_t^p(x, \zeta; F) := \underset{y_{0:p}, v_{0:p-1}}{\arg\min} \sum_{\tau=1}^p f_{t+\tau}(y_\tau) + \sum_{\tau=1}^p c_{t+\tau}(v_{\tau-1}) + F(y_k)$$
$$\text{s.t. } y_\tau = A_{t+\tau-1} y_{\tau-1} + B_{t+\tau-1} v_{\tau-1} + \zeta_{\tau-1}, \tau = 1, \ldots, p, \qquad (2)$$
$$y_0 = x,$$

where $x \in \mathbb{R}^n$ is the initial state, $\zeta \in (\mathbb{R}^n)^p$ (indexed by $0, \ldots, p - 1$) is a sequence of disturbances, and $F : \mathbb{R}^n \to \mathbb{R}$ is a standard terminal cost function regularizing the final state. Here we additionally require that the terminal cost $F$ has the form $F(x) = \alpha(\|x\|)$, where $\alpha : \mathbb{R}_{\geq 0} \to \mathbb{R}_{\geq 0}$ is a convex K-function (i.e. continuous increasing function with 0 at the origin, see Appendix A for definition) that is twice continuously differentiable. For each time step $\tau = 1, \ldots, k$, $y_\tau \in \mathbb{R}^n$ is the predictive state, and $v_\tau \in \mathbb{R}^m$ is the predictive control action. To make the algorithm well-defined, at time step $t = T - k$, $PC_k$ can finish the rest of the trajectory optimally by committing $u_{T-k:T-1} = \tilde{\psi}(x_{T-k}, w_{T-k:T-1}; 0)$. The pseudo-code of predictive control is given in Algorithm 1.

It is also important in our framework to study the behavior of predictive control under some fixed terminal point. So, for prediction length $p \geq 1$ and time step $t$, we define an auxiliary optimization problem with a strict terminal constraint $y_p = z$ as follows:

$$\psi_t^p(x, \zeta, z) := \underset{y_{0:p}, v_{0:p-1}}{\arg\min} \sum_{\tau=1}^p f_{t+\tau}(y_\tau) + \sum_{\tau=1}^p c_{t+\tau}(v_{\tau-1})$$
$$\text{s.t. } y_\tau = A_{t+\tau-1} y_{\tau-1} + B_{t+\tau-1} v_{\tau-1} + \zeta_{\tau-1}, \tau = 1, \ldots, p, \qquad (3)$$
$$y_0 = x, y_p = z,$$

where the optimal value is denoted by $\iota_t^p(x, \zeta, z)$. We define this auxiliary optimization problem besides $\tilde{\psi}_t^p(x, \zeta; F)$ because we need to fix both the initial state and the terminal state, for example, when expressing a sub-trajectory of the offline optimal trajectory as the solution of an optimization problem. $\psi$ also allows us to study the impact of the perturbation at the terminal state on the optimal trajectory directly, which will be useful in the proof of dynamic regret (Theorem 4.2) and competitive ratio (Theorem 4.3).

Throughout the paper, we use $\{(x_t, u_t)\}_{t=1}^T$ to denote the trajectory of predictive control, and use $\{(x_t^*, u_t^*)\}_{t=1}^T$ to denote the offline optimal trajectory (i.e., the optimal solution of (1)). We also use several standard definitions and notations in linear algebra and optimization, which we detail in Appendix A for clarity. In particular, we use vector 2-norms and induced matrix 2-norms throughout this paper unless otherwise specified.

Two standard metrics will be utilized to assess the performance of our predictive control algorithms, namely *dynamic regret* $\sup_{x(0), w} \big( \text{cost}(ALG) - \text{cost}(OPT) \big)$ (the additional cost of our algorithm against the optimal algorithm) and *competitive ratio* $\sup_{x(0), w} \frac{\text{cost}(ALG)}{\text{cost}(OPT)}$ (the worst-case ratio of the cost of our algorithm over that of the optimal algorithm).

# 3 A Perturbation Approach

In order to study the regret and competitive ratio of controllers in LTV systems, we develop a new analysis based on a perturbation approach, which we introduce in this section. This approach is based on developing bounds on how much the solutions to (2) and (3) change with respect to perturbations to the initial/terminal states and the disturbance sequence. Our perturbation bounds are related to the concept of *incremental stability* defined in [34], but not exactly the same because we consider the optimal trajectory in a finite horizon whereas the incremental stability focuses on asymptotic behavior over an infinite horizon. Simply stated, the key to our approach is to derive the perturbation bound in Theorem 3.3, which states that if the target variable we are concerned with is the $h$-th predictive state/control input, while the perturbation occurs at the $\tau$-th time step, then the impact on the target variable is exponentially small with respect to the time difference $|h - \tau|$.

Proving such a result directly is challenging because of the complexity of the LTV dynamical constraints in (2) and (3). Thus, we develop a novel reduction from LTV systems to fully-actuated systems, i.e., systems where the controller can steer the system to any state in the whole space $\mathbb{R}^n$ freely at every time step. This special case is a form of online optimization called *smoothed online convex optimization* (SOCO), and has received considerable attention recently, e.g., [24, 27, 35]. We exploit the controllability of the dynamics to analyze the LTV system in chunks of $d$ time steps. A sequence of $d$ time steps combined together can be thought as a fully-actuated system and thus we can formulate a SOCO problem, which is $(1/d)$-times as long as the original LTV system. In this section, we first show the perturbation bound for SOCO in Section 3.1, and then we leverage our reduction to derive a result for general LTV systems in Section 3.2.

## 3.1 Smoothed Online Convex Optimization

The classic setting of SOCO is an online game played by an agent against an adversary: at each time step $t$, the adversary reveals a hitting cost function $\hat{f}_t$, a switching cost function $\hat{c}_t$, and a disturbance (or exogenous input) $\hat{w}_t$. The agent picks a decision point $\hat{x}_t \in \mathbb{R}^n$, and incurs a stage cost of $\hat{f}_t(\hat{x}_t) + \hat{c}_t(\hat{x}_t, \hat{x}_{t-1}, \hat{w}_{t-1})$. The agent seeks to minimize the total cost it incurs throughout the game. The offline optimal cost is defined as the minimum cost if the agent has full knowledge of the costs and disturbances at the start of the game. Instead of analyzing the performance of an online algorithm directly, our focus is on studying how the perturbations of the system parameters (initial state, terminal state, and disturbances) impact the offline optimal solution. These results are critical for deriving the guarantees for predictive control in the online setting in Section 4.

To begin, observe that when the initial state $\hat{x}_0$, terminal state $\hat{x}_p$, and the disturbances $\hat{w}$ are given, the optimal $p$-step trajectory of SOCO can be obtained from the unconstrained optimization problem

$$\hat{\psi}(\hat{x}_0, \hat{w}, \hat{x}_p) := \arg\min_{\hat{x}_{1:p-1}} \sum_{\tau=1}^{p-1} \hat{f}_\tau(\hat{x}_\tau) + \sum_{\tau=1}^{p} \hat{c}_\tau(\hat{x}_\tau, \hat{x}_{\tau-1}, \hat{w}_{\tau-1}), \tag{4}$$

where the objective is a convex function of the decision variables $\hat{x}_{1:p-1}$. Since (4) is an unconstrained optimization problem, the gradient of its objective equals zero at $\hat{\psi}(\hat{x}_0, \hat{w}, \hat{x}_p)$. Using this, we can further show that the directional derivative of $\hat{\psi}(\hat{x}_0, \hat{w}, \hat{x}_p)$ along some direction $e$, denoted by $\chi$, satisfies the linear equation $M\chi = \delta$, where symmetric matrix $M$ is the Hessian of the objective and vector $\delta$ is determined by the direction $e$. A special structure of the objective of (4) is that the correlations only occur in two consecutive time steps. This implies that its Hessian $M$ is block

tri-diagonal. Such tri-diagonal structure of $M$ has been noted by previous work, e.g. [36], and have been leveraged to solve the linear equation $M\chi = \delta$ quickly. In contrast, we focus on the exponential decay phenomena $M^{-1}$ exhibits, i.e., the magnitudes of entries decay exponentially with respect to their distances to the main diagonal [37]. Bounding each entry of $\chi = M^{-1}\delta$ separately gives us the following perturbation bound. We state this result formally in Theorem 3.1, and its proof can be found in Appendix B.

**Theorem 3.1.** *Given a tuple $(\hat{x}_0, \hat{w}, \hat{x}_p)$ that contains the initial state, the disturbances, and the terminal state in this order, we consider the optimal solution of the SOCO problem*

$$\hat{\psi}(\hat{x}_0, \hat{w}, \hat{x}_p) := \arg\min_{\hat{x}_{1:p-1}} \sum_{\tau=1}^{p-1} \hat{f}_\tau(\hat{x}_\tau) + \sum_{\tau=1}^{p} \hat{c}_\tau(\hat{x}_\tau, \hat{x}_{\tau-1}, \hat{w}_{\tau-1})$$

*indexed by $1, \ldots, p-1$. Assume $\hat{f}_\tau : \mathbb{R}^n \to \mathbb{R}$ is $\mu$-strongly convex, $\hat{c}_\tau : \mathbb{R}^n \times \mathbb{R}^n \times \mathbb{R}^r \to \mathbb{R}$ is convex and $\ell$-strongly smooth, and both are twice continuously differentiable for $\tau = 1, \ldots, p$, then*

$$\left\| \hat{\psi}(\hat{x}_0, \hat{w}, \hat{x}_p)_h - \hat{\psi}(\hat{x}'_0, \hat{w}', \hat{x}'_p)_h \right\| \le C_0 \left( \lambda_0^{h-1} \|\hat{x}_0 - \hat{x}'_0\| + \sum_{\tau=0}^{p-1} \lambda_0^{|h-\tau|-1} \|\hat{w}_\tau - \hat{w}'_\tau\| + \lambda_0^{p-h-1} \|\hat{x}_p - \hat{x}'_p\| \right)$$

*for all $1 \le h \le p-1$, where $C_0 = (2\ell)/\mu$ and $\lambda_0 = 1 - 2 \cdot \left( \sqrt{1 + (2\ell/\mu)} + 1 \right)^{-1}$.*

As a remark, we do not require the hitting cost $\hat{f}_\tau$ to be strongly smooth, or the switching cost $\hat{c}_\tau$ to be strongly convex in Theorem 3.1. This makes the assumptions on the SOCO costs $\hat{f}_\tau, \hat{c}_\tau$ weaker than the assumptions on the LTV costs $f_\tau, c_\tau$ defined in (1).

## 3.2 Linear Time-Varying System

We now build upon the SOCO perturbation result to derive a perturbation result for LTV systems. In particular, we show an exponentially-decaying perturbation bound for our LTV system by reducing it to SOCO and apply Theorem 3.1. As we have discussed, LTV systems are more difficult than SOCO because the dynamics prevent the online agent from picking the next state $x_{t+1}$ freely at a given state $x_t$. We overcome this obstacle by redefining the decision points as illustrated in Figure 1. Specifically, given state $x_t$ at time step $t$ as the last decision point, we then ask the online agent to decide state $x_{t+d}$ at time step $(t+d)$ rather than $x_{t+1}$ at time step $(t+1)$.

Since $d$ is the controllability index, $x_{t+d}$ can be picked freely from the whole space $\mathbb{R}^n$ regardless of $x_t$. We also utilize the *principle of optimality*, e.g. if $y_{0:k}, v_{0:k-1}$ is the optimal solution to $\psi_t^k(x, \xi, z)$, then $y_{i:j}, v_{i:j-1}$ is the optimal solution to $\psi_{t+i}^{j-i}(y_i, \xi_{i:j-1}, y_j)$ for any $0 \le i < j \le k$. Therefore, the trajectory between time $t$ and $(t+d)$ can be recovered by solving $\psi_t^d(x_t, w_{t:t+d-1}, x_{t+d})$. So we are able to formulate a valid SOCO problem on the sequence of time steps $t, t+d, t+2d, \ldots$.

Naturally, the hitting cost at time step $(t+d)$ remains the same, while the switching cost becomes $\xi_t^d(x_t, w_{t:t+d-1}, x_{t+d})$, where the function $\xi_t^p$ is defined as

$$\xi_t^p(x, \zeta, z) := \iota_t^p(x, \zeta, z) - f_{t+p}(z). \tag{5}$$

An illustration of the reduction can be found in Figure 1. We would like to point out that our reduction from optimal control to SOCO is novel in that it leverages the principle of optimality to apply to more general LTV settings, as opposed to the reduction via control canonical forms in [3] that is specific to LTI systems. Unlike the switching costs in [14, 27, 35, 38] which are explicitly defined as the $\ell_2$-distance or squared $\ell_2$-distance, the switching cost $\xi_t^p$ here is defined implicitly as the optimal value of an optimization problem. Lemma 3.2 shows that the switching cost defined in (5) satisfies the requirements of Theorem 3.1, which allows us to obtain the desired perturbation bound.

**Lemma 3.2.** *Under Assumption 2.1 and 2.2, for integer $p \ge d$, we have*

1. *$\psi_t^p(x, \zeta, z)$ is $L_1(p)$-Lipschitz in $(x, \zeta, z)$;*

2. *$\xi_t^p(x, \zeta, z)$ is convex and $L_2(p)$-strongly smooth in $(x, \zeta, z)$.*

*Here $L_1(p) = C(p)(1 + \ell \cdot C(p)/m_c), L_2(p) = \ell \cdot C(p)^2 + \ell^2 \cdot C(p)^4/m_c$, where $\ell = \max(\ell_f, \ell_c)$,*

$$C(p) = \begin{cases} O(a^{3p}) & \text{if } a > 1; \\ O(p^2) & \text{if } a = 1; \\ O(1) & \text{if } a < 1. \end{cases}$$

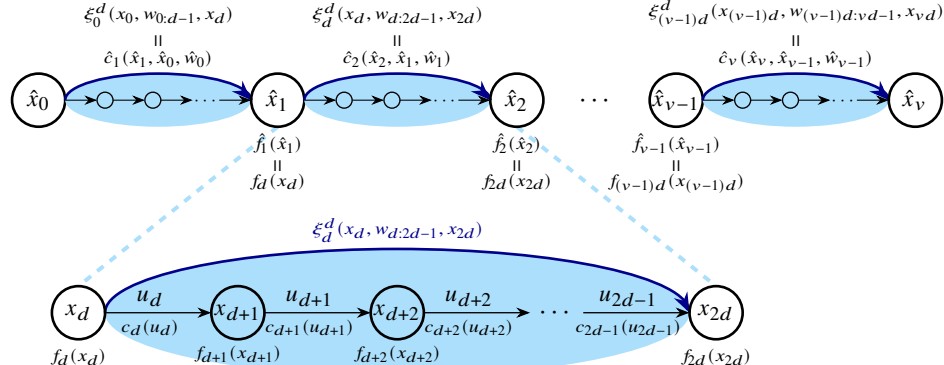

Figure 1: Illustration of the reduction from LTV to SOCO. Here we consider a simple example where $t = 0$ and $p = vd$. At time step 0, the agent cannot steer the system to an arbitrary target state at the next time step due to dynamical constraints. However, given $(d, \sigma)$-uniform controllability, the controller is able to enforce an arbitrary target state after $d$ time steps, which prompts the transformation to a SOCO problem with a decision point in every $d$ time steps.

In Lemma 3.2, we use $O(\cdot)$ to hide quantities $a, b,$ and $1/\sigma$; the precise expression of $C(p)$ and the proof of Lemma 3.2 can be found in Appendix C. Using the reduction from LTV to SOCO, we obtain a perturbation bound for the LTV systems (2) and (3) in Theorem 3.3, the proof of which is deferred to Appendix D.

**Theorem 3.3.** *Consider the optimization problem defined in (2) and (3) and with a horizon length $p \geq d$. Under Assumptions 2.1 and 2.2, given any $(x, \zeta, z)$ and $(x', \zeta', z')$,*

$$\left\| \tilde{\psi}_t^p(x, \zeta; F)_{y_h} - \tilde{\psi}_t^p(x', \zeta'; F)_{y_h} \right\| \leq C \left( \lambda^h \|x - x'\| + \sum_{\tau=0}^{p-1} \lambda^{|h-\tau|} \|\zeta_\tau - \zeta_\tau'\| \right)$$

$$\left\| \psi_t^p(x, \zeta, z)_{y_h} - \psi_t^p(x', \zeta', z')_{y_h} \right\| \leq C \left( \lambda^h \|x - x'\| + \sum_{\tau=0}^{p-1} \lambda^{|h-\tau|} \|\zeta_\tau - \zeta_\tau'\| + \lambda^{p-h} \|z - z'\| \right)$$

*hold for all time steps $t$. Here we define $L_0 = \max_{d \leq p \leq 2d-1} L_2(p)$, and the constants are given by*

$$\lambda = \left( 1 - 2 \left( \sqrt{1 + (2L_0/m_c)} + 1 \right)^{-1} \right)^{\frac{1}{2d-1}}, C = \frac{2L_0}{m_c} \cdot \left( 1 - 2 \left( \sqrt{1 + (2L_0/m_c)} + 1 \right)^{-1} \right)^{-1}.$$

Theorem 3.3 allows us to bound the distance between any two trajectories so long as they can be expressed as the optimal solutions of the optimization problem (2) or (3). For example, to bound the norm of each state in the predictive trajectory $\tilde{\psi}_t^p(x, \zeta; F)$, we only need to set $x' = 0, \zeta' = 0$ in the first inequality because an all zero trajectory can be expressed as $\tilde{\psi}_t^p(0, 0; F)$. The formal statement of this result can be found in Appendix E.

## 4 Performance Guarantees for Predictive Control

We now demonstrate the power of the perturbation approach in Section 3.2 by obtaining bounds on regret and competitive ratio. The key intuition behind our analysis is the following: at time step $t$, if the predictive controller with prediction window $k$ is given the knowledge of $x_t^*$ and $x_{t+k}^*$, it can fully recover the offline optimal states and control inputs for the future $k$ time steps, $x_{t+1:t+k}^*$ and $u_{t:t+k-1}^*$, from $\psi_t^k(x_t^*, w_{t:t+k-1}, x_{t+k}^*)$. However, without the knowledge of the offline optimal states, the predictive controller solves $\psi_t^k(x_t, w_{t:t+k-1}, x_{t+k})$ instead, where $x_{t+k}$ is implicitly determined by the $k$-th predictive state of $\tilde{\psi}_t^k(x_t, w_{t:t+k-1}; F)$. We overcome this gap with our perturbation approach (specifically, Theorem 3.3 and its corollaries), which allows us to bound the distance between the controller's trajectory and the offline optimal trajectory.

## 4.1 Dynamic Regret

We first bound the dynamic regret of predictive control. For this analysis, a key observation is that the offline optimal trajectory is given by $x^* = \tilde{\psi}_0^T(x_0, w_{0:T-1}; 0)_{y_{1:T}}$. Furthermore, the optimal trajectory starting at time step $t$ with state $x_t$ is equivalent to the trajectory of predictive control with prediction window $(T-t)$ and no terminal cost, i.e. $\tilde{\psi}_t^{T-t}(x_t, w_{t:T-1}; 0)_{y_{1:T-t}}$. Using Theorem 3.3, we can bound the change in decision points against the change in prediction window $k$. Lemma 4.1 formalizes this:

**Lemma 4.1.** *For any integers $p, h$ such that $p \geq h \geq 1$ and time step $t < T - p$, we have*

$$\left\| \tilde{\psi}_t^p(x_t, w_{t:t+p-1}; F)_{y_h} - \tilde{\psi}_t^{p+1}(x_t, w_{t:t+p}; F)_{y_h} \right\| \leq 2C\lambda^{p-h}\left( C\lambda^p \|x_t\| + \frac{2C}{1-\lambda} \sup_{0 \leq \tau \leq T-1} \|w_\tau\| \right).$$

Then, we can follow the road map below to bound the dynamic regret of predictive control $PC_k$:

(a) Given the well-conditioned state/control costs, it suffices to bound the distance between $PC_k$'s trajectory and the offline optimal trajectory (i.e., $\|x_t - x_t^*\|$) to show the dynamic regret result. See inequalities (22) and (23) in Appendix H for technical details.

(b) At each time step $t$, the optimal next state (under an imaginary terminal cost $F$) from the current state $x_t$ is given by $\tilde{\psi}_t^{T-t}(x_t, w_{t:T-1}; F)_{y_1}$. However, reaching the optimal next state from $x_t$ requires full knowledge of the future costs, dynamics, and disturbances. Although $PC_k$ cannot reach the optimal next state due to incomplete knowledge of the future, it can leverage the predictions of future $k$ steps to decide a near-optimal control action from state $x_t$. By cumulatively summing up the bounded difference in Lemma 4.1 and applying Theorem 3.3, one can show the suboptimality, measured by the distance $\left\| x_{t+1} - \tilde{\psi}_t^{T-t}(x_t, w_{t:T-1}; F)_{y_1} \right\|$, is in the order of $O(\lambda^k)$. See inequality (19) in Appendix H for technical details.

(c) Using the exponentially-decaying LTV perturbation bound in Theorem 3.3, we can convert the per-step suboptimality bounds to a global suboptimality bound on $\|x_t - x_t^*\|$ that is also in the order of $O(\lambda^k)$. See inequalities (20) and (21) in Appendix H for technical details.

The $O(\lambda^k)$ upper bound on the distance between the algorithm's trajectory and the offline optimal trajectory leads to the regret bound in Theorem 4.2.

**Theorem 4.2.** *Suppose $\|w_t\| \leq D$ for some constant $D$ at each time step $t$. Let $\lambda, C, L_0$ be the decay rate and constants defined in Theorem 3.3. If prediction window $k \geq d$ is sufficiently large, such that*

$$k \geq 1 + \log\left( \frac{1}{1-\delta} \cdot C\left( \frac{2C}{1-\lambda} + \lambda \right) \right) \Big/ \log\left( \frac{1}{\lambda} \right) \tag{6}$$

*for some positive constant $\delta \in (0, 1)$, then the trajectory of $PC_k$ satisfies:*

1. *(Input-to-state Stability) The norm of each state $x_t$ is upper bounded by*

$$\|x_t\| \leq \begin{cases} \frac{C}{\delta} \cdot (1-\delta)^{\max(0, t-k)} \|x_0\| + \frac{2C}{\delta(1-\lambda)}\left( 1 + \frac{2C}{1-\lambda} \right) D & \text{if } 0 < t \leq T - k \\ \frac{C^2}{\delta} \cdot (1-\delta)^{T-2k} \lambda^{t+k-T} \|x_0\| + \left( \frac{2C^2}{\delta(1-\lambda)}\left( 1 + \frac{2C}{1-\lambda} \right) + \frac{2C}{1-\lambda} \right) D & \text{if } T - k < t \leq T. \end{cases}$$

2. *(Dynamic Regret) The dynamic regret of $PC_k$ is upper bounded by*

$$\text{cost}(PC_k) - \text{cost}(OPT) = O\left( \left( D + \frac{\lambda^k(\|x_0\| + D)}{\delta} \right)^2 \lambda^k T + \lambda^k \|x_0\|^2 \right),$$

*where the notation hides quantities $a, b', \ell_f, \ell_c, C, 1/(1-\lambda)$ and $L_0$.*

An implication of Theorem 4.2 is that to obtain $o(1)$ dynamic regret when the norm of disturbances are uniformly upper bounded, it suffices to use a prediction window of length $\Theta(\log T)$. This parallels the result shown in [4], although in a more general setting.

## 4.2 Competitive Ratio

We now focus on bounding the competitive ratio of predictive control. Here, we study a modification of the predictive control algorithm we have considered to this point. In particular, we introduce a

---

**Algorithm 2** Predictive Control with Replan Window $h$ ($PC_{(k,h)}$)

---
1: Suppose $T = n_0 h + m_0$, where integers $n_0 \geq 0$, $k - h + 1 \leq m_0 \leq k$.
2: **for** $t = 0, h, \ldots, n_0(h-1)$ **do**
3:      Observe current state $x_t$ and receive predictions $\vartheta_{t:t+k-1}$.
4:      Solve and commit control actions $u_{t:t+h-1} := \tilde{\psi}_t^k(x_t, w_{t:t+k-1}; F)_{v_{0:h-1}}$.
5: At time step $t = n_0 h$, observe current state $x_t$ and receive predictions $\vartheta_{t:T-1}$.
6: Solve and commit control actions $u_{t:T-1} := \tilde{\psi}_t^{m_0}(x_t, w_{t:T-1}; 0)_{v_{0:m_0-1}}$.

---

replan window $h$, as defined in Algorithm 2 which we denote as $PC_{(k,h)}$. This style of algorithm has been considered previously in the SOCO literature, where it has been shown to obtain a constant competitive ratio in some settings where MPC does not [39].

Our analysis approach highlights why this modification is beneficial for competitive ratio. Specifically, we obtain the competitive ratio bound by applying a potential method building on [40]. We define the potential function as the squared distance between the algorithm's trajectory and the offline optimal trajectory, i.e., $\phi_t(x_t, x_t^*) = \left\| x_t - x_t^* \right\|^2$, which is standard in the literature [5, 14, 27]. We study how this potential function changes over time. Intuitively, we need to upper bound the increment of this potential function by the offline optimal cost to obtain a competitive ratio result. To achieve this, the algorithm needs to "move closer" to the offline optimal trajectory rather than "moving further away" from it. Recall that Theorem 3.3 gives that

$$\left\| \psi_t^k(x_t, w_{t:t+k-1}; F)_{y_h} - \psi_t^k(x_t^*, w_{t:t+k-1}; F)_{y_h} \right\| \leq C\lambda^h \left\| x_t - x_t^* \right\|. \tag{7}$$

When the algorithm commits the first predictive state ($h = 1$), the left hand side of (7) might be larger than $\left\| x_t - x_t^* \right\|$ when $C\lambda > 1$. Thus, the algorithm must "wait" until the right hand side of (7) becomes smaller than $\left\| x_t - x_t^* \right\|$. This is accomplished in Algorithm 2 via the replan window $h$.

Our main result for this section is the following competitive ratio bound for $PC_{(k,h)}$.

**Theorem 4.3.** *Let $\lambda, C, L_0$ be the decay rate and constants defined in Theorem 3.3. In Algorithm 2, if the replan window $h$ satisfies $h \geq \max\{\log((1+\varepsilon)C)/\log(1/\lambda), d\}$ for some positive constant $\varepsilon$, and the prediction window $k$ satisfies $k \geq h + d$, then it has competitive ratio*

$$\sup_{x(0),w} \frac{\mathrm{cost}(PC_{(k,h)})}{\mathrm{cost}(OPT)} = 1 + O\left( \varepsilon^{-1} \left( \frac{L_0 + \ell_f}{m_f} \right)^{1/2} \cdot C\lambda^{k-1-h} \right),$$

*where the notation only hides a small numerical constant.*

Note that when the constant $\varepsilon$ and the replan window $h$ are fixed, the competitive ratio is on the order of $1 + O(\rho^k)$ as the length of prediction $k$ tends to infinity. One potential line of future work is to understand if the replan window is necessary. It may be possible to either strengthen the constants given in Theorem 3.3 or improve our proof approach so as to eliminate the requirement on $h$.

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
