# A  Definitions and Notations

**Definition A.1.** *We use the follow convention on linear algebra:*

1. *$\|\cdot\|$ denotes the (Euclidean) 2-norm for vectors and the induced 2-norm for matrices:*

$$\|v\| = \sqrt{v_1^2 + v_2^2 + \cdots + v_n^2}, \ v \in \mathbb{R}^n$$

$$\|A\| = \sup_{v \in \mathbb{R}^n \setminus \{0\}} \frac{\|Ax\|}{\|x\|}, \ A \in \mathbb{R}^{m \times n};$$

2. *$\sigma(A)$ is the collection of singular values of a matrix $A$, also known as the singular spectrum;*

3. *$\sigma_{\min}(A)$ denotes the smallest singular value of a matrix $A$;*

4. *$A \geq 0$ indicates that a matrix $A$ is positive semi-definite.*

The notions of strong-convexity and smoothness are used throughout this paper:

**Definition A.2.** *A real-valued function $g : \mathbb{R}^n \to \mathbb{R}$ is called $\ell$-strongly smooth if*

$$g(y) \leq g(x) + \langle \nabla g(x), y - x \rangle + \frac{\ell}{2} \|y - x\|_2^2$$

*and is called m-strongly convex if*

$$g(y) \geq g(x) + \langle \nabla g(x), y - x \rangle + \frac{m}{2} \|y - x\|_2^2$$

*for any $x, y \in \mathbb{R}^n$. Here $\langle \cdot, \cdot \rangle$ denotes the standard inner product of vectors.*

We also require the terminal cost to be a K-function, the definition of which is given below.

**Definition A.3.** *A function $g : \mathbb{R}_{\geq 0} \to \mathbb{R}_{\geq 0}$ is said to be a K-function (or belongs to class K), if it is continuous, strictly increasing, and satisfies $g(0) = 0$.*

For ease of reference, we summarize in the following table all the notation used in the paper.

| Notation | Meaning |
|---|---|
| $A_t, B_t$ | dynamical matrices of the system at time step $t$ |
| $x_t, u_t, w_t$ | state, control action, and disturbance at time step $t$ |
| $f_t, c_t$ | hitting cost function and control cost function at time step $t$ |
| $m_f, m_c$ | strong convexity parameters of cost functions |
| $\ell_f, \ell_c$ | strong smoothness parameters of cost functions |
| $\tilde{\psi}_t^p(x, \zeta; F)$ | optimal trajectory[†] from step $t$ in the future $p$ steps (free terminal state) |
| $\tilde{\iota}_t^p(x, \zeta; F)$ | optimal value from step $t$ in the future $p$ steps (free terminal state) |
| $\psi_t^p(x, \zeta, z)$ | optimal trajectory[†] from step $t$ in the future $p$ steps (fixed terminal state $z$) |
| $\iota_t^p(x, \zeta, z)$ | optimal value from step $t$ in the future $p$ steps (fixed terminal state $z$) |
| $y_\tau, v_\tau, \zeta_\tau$ | (in $\psi$ and $\tilde{\psi}$) predictive state, control action, and disturbance within the optimization |
| $F, z$ | terminal constraint function $F$ (in $\psi$) or fixed terminal state $z$ (in $\tilde{\psi}$) |
| $\{(x_t, u_t)\}_{t=1}^T$ | the trajectory of our predictive control algorithm |
| $\{(x_t^*, u_t^*)\}_{t=1}^T$ | the offline optimal trajectory (i.e., $\tilde{\psi}_0^T(x(0), w; 0)$) |
| $\hat{\psi}(\hat{x}_0, \hat{w}, \hat{x}_p)$ | the converted SOCO trajectory with initial state $\hat{x}_0$, disturbances $\hat{w}$, and terminal state $\hat{x}_p$ |
| $\hat{f}_\tau, \hat{c}_\tau, \hat{x}_{1:p-1}$ | hitting cost, transition cost, and optimal trajectory of the converted SOCO problem |

[†] Trajectory contains both states and control inputs, which are referred to by subscripts $y_\tau$ and $u_\tau$, respectively.

# B  Proof of Theorem 3.1

In the next lemma we will use the notation $A_{S_R, S_C}$ to denote the submatrix obtained by selecting the blocks indexed by some set $S_R \times S_C$ while preserving their relative order. Specifically, consider a

matrix $A \in \mathbb{R}^{\omega n \times \omega n}$ formed by $\omega \times \omega$ blocks $A_{i,j} \in \mathbb{R}^{n \times n}$. Let $i_1 < \cdots < i_{|S_R|}$ be the elements in $S_R \subseteq \{1, \ldots, \omega\}$, and $j_1 < \cdots < j_{|S_C|}$ be the elements in $S_C \subseteq \{1, \ldots, \omega\}$, both in ascending order. Then $A_{S_R,S_C} \in \mathbb{R}^{|S_R|n \times |S_C|n}$ is defined as a block matrix

$$
A_{S_R,S_C} := \begin{bmatrix}
A_{i_1,j_1} & A_{i_1,j_2} & \cdots & A_{i_1,j_{|S_C|}} \\
A_{i_2,j_1} & A_{i_2,j_2} & \cdots & A_{i_2,j_{|S_C|}} \\
\vdots & \vdots & \ddots & \vdots \\
A_{i_{|S_R|},j_1} & A_{i_{|S_R|},j_2} & \cdots & A_{i_{|S_R|},j_{|S_C|}}
\end{bmatrix}.
$$

For a diagonal block matrix $D = \mathrm{diag}(D_1, \ldots, D_\omega)$ and a set $S \subseteq \{1, \ldots, \omega\}$, we use the shorthand notation $D_S := \mathrm{diag}\left(D_{i_1}, D_{i_2}, \ldots, D_{i_{|S|}}\right)$, where $i_1 < \ldots < i_{|S|}$ are the elements in $S$.

**Lemma B.1.** *Suppose $A$ is a positive definite matrix in $\mathbb{S}^{\omega n}$ formed by $\omega \times \omega$ blocks $A_{i,j} \in \mathbb{R}^{n \times n}$. Assume that $A$ is $q$-banded for an even positive integer $q$; that is*

$$
A_{i,j} = 0, \forall |i - j| > q/2.
$$

*Let $[a_0, b_0]$ ($b_0 > a_0 > 0$) be the smallest interval containing the spectrum $\sigma(A)$. Suppose $D = \mathrm{diag}(D_1, \ldots, D_\omega)$, where $D_i \in \mathbb{S}^n$ is positive semi-definite. Let $M = \left((A + D)^{-1}\right)_{S_R,S_C}$ as defined above, where $S_R, S_C \subseteq \{1, \ldots, \omega\}$. Then we have $\|M\| \leq C\gamma^{\hat{d}}$, where*

$$
C = \frac{2}{a_0}, \gamma = \left(\frac{\sqrt{\mathrm{cond}(A)} - 1}{\sqrt{\mathrm{cond}(A)} + 1}\right)^{2/q}, \hat{d} = \min_{i \in S_R, j \in S_c} |i - j|.
$$

*Here $\mathrm{cond}(A) = b_0/a_0$ denotes the condition number of matrix $A$.*

*Proof of Lemma B.1.* We first prove the lemma for the the special case where $D = 0$.

For the case $\hat{d} \neq 0$, write $\hat{d} = \upsilon q/2 + \kappa$ for integers $\upsilon, \kappa$ satisfying $\upsilon \geq 0, 1 \leq \kappa \leq q/2$. Following the same approach as the proof of Proposition 2.2 in [37], we see that there exists a polynomial $p_\upsilon$ of degree $\upsilon$, where

$$
\left\|A^{-1} - p_\upsilon(A)\right\| \leq \frac{1}{a_0} \cdot \frac{\left(1 + \sqrt{\mathrm{cond}(A)}\right)^2}{2\,\mathrm{cond}(A)}\gamma^{\hat{d}} \leq C\gamma^{\hat{d}},
$$

where the last inequality holds because $\mathrm{cond}(A) \geq 1$.

Since $p_v$ has degree $v < \frac{2\hat{d}}{q}$ and $A$ is $q$-banded, the matrix $p_\upsilon(A)$ satisfies $(p_\upsilon(A))_{i,j} = 0$ for any $i \in S_R$ and $j \in S_C$. We then obtain

$$
\|P\| = \left\|\left(A^{-1}\right)_{S_R,S_C}\right\| = \left\|\left(A^{-1} - p_\upsilon(A)\right)_{S_R,S_C}\right\| \leq \left\|A^{-1} - p_\upsilon(A)\right\| \leq C\gamma^{\hat{d}},
$$

because 2-norm of a submatrix cannot be larger than that of the original matrix.

For the case $\hat{d} = 0$, as $\|P\| = \left\|\left(A^{-1}\right)_{S_R,S_C}\right\| \leq \|A^{-1}\| = \frac{1}{a_0} \leq C$, the result trivially holds.

Now we show the general case (where $D_i \geq 0$ for $1 \leq i \leq n$) through a reduction to the special case. Define a positive definite matrix $N := (a_0 I + D) \in \mathbb{S}^{n\omega}$, and then define matrix $H \in \mathbb{S}^{n\omega}$ as follows,

$$
H = N^{-\frac{1}{2}}(A + D)N^{-\frac{1}{2}}.
$$

We start by showing that $I \leq H \leq \frac{b_0}{a_0} \cdot I$. For any $x \in \mathbb{R}^{n\omega}$, we observe

$$
\begin{aligned}
x^\top H x &= x^\top N^{-\frac{1}{2}} A N^{-\frac{1}{2}} x + x^\top N^{-\frac{1}{2}} D N^{-\frac{1}{2}} x \\
&\geq x^\top N^{-\frac{1}{2}} a_0 I N^{-\frac{1}{2}} x + x^\top N^{-\frac{1}{2}} D N^{-\frac{1}{2}} x \\
&= x^\top N^{-\frac{1}{2}} (a_0 I + D) N^{-\frac{1}{2}} x \\
&= \|x\|^2.
\end{aligned}
$$

For the other inequality, we have

$$
\begin{aligned}
x^\top H x &= x^\top N^{-\frac{1}{2}} A N^{-\frac{1}{2}} x + x^\top N^{-\frac{1}{2}} D N^{-\frac{1}{2}} x \\
&\leq x^\top N^{-\frac{1}{2}} b_0 I N^{-\frac{1}{2}} x + x^\top N^{-\frac{1}{2}} D N^{-\frac{1}{2}} x \\
&= x^\top N^{-\frac{1}{2}} (a_0 I + D) N^{-\frac{1}{2}} x + (b_0 - a_0) x^\top N^{-1} x \\
&\leq \|x\|^2 + \frac{b_0 - a_0}{a_0} \cdot \|x\|^2 \\
&= \frac{b_0}{a_0} \cdot \|x\|^2.
\end{aligned}
$$

Thus $I \leq H \leq \frac{b_0}{a_0} \cdot I$, which gives $\mathrm{cond}(H) \leq \frac{b_0}{a_0} = \mathrm{cond}(A)$. Note that $H$ is also $q$-banded, so we can apply the result of the special case ($D_i = 0, i = 1, \cdots, n$) to obtain that

$$
\left\| (H^{-1})_{S_R, S_C} \right\| \leq 2 \gamma_H^{\hat{d}} \leq 2 \gamma^{\hat{d}},
$$

where $\gamma_H = \left( \frac{\sqrt{\mathrm{cond}(H)} - 1}{\sqrt{\mathrm{cond}(H)} + 1} \right)^{2/q} \leq \gamma$. Using this inequality, we conclude that

$$
\begin{aligned}
\|P\| = \left\| ((A + D)^{-1})_{S_R, S_C} \right\| &= \left\| \left( N^{-\frac{1}{2}} H^{-1} N^{-\frac{1}{2}} \right)_{S_R, S_C} \right\| \\
&\leq \left\| (a_0 I + D_{S_R})^{-\frac{1}{2}} \right\| \cdot \left\| (H^{-1})_{S_R, S_C} \right\| \cdot \left\| (a_0 I + D_{S_C})^{-\frac{1}{2}} \right\| \\
&\leq \frac{1}{a_0} \left\| (H^{-1})_{S_R, S_C} \right\| \\
&\leq C \gamma^{\hat{d}}.
\end{aligned}
$$

Here we apply the fact that $\left\| (a_0 I + D_S)^{-\frac{1}{2}} \right\| \leq \frac{1}{\sqrt{a_0}}$ since $D_S \geq 0$. $\qquad \square$

Now we return to the proof of Theorem 3.1

*Proof of Theorem 3.1.* Let $e = (e_0^\top, \mu^\top, e_p^\top)^\top$ be a vector where $e_0, e_p \in \mathbb{R}^n$ and

$$
\mu = [\mu_0, \mu_1, \ldots, \mu_{p-1}],
$$

for $\mu_i \in \mathbb{R}^r, i = 0, 1, \ldots, p - 1$. Let $\theta$ be an arbitrary real number. Define function $\hat{h} : \mathbb{R}^{(p-1) \times n} \times \mathbb{R}^n \times \mathbb{R}^{p \times r} \times \mathbb{R}^n \to \mathbb{R}_+$ as

$$
\hat{h}(\hat{x}_{1:p-1}, \hat{x}_0, \hat{w}_{0:p-1}, \hat{x}_p) = \sum_{\tau=1}^{p-1} \hat{f}_\tau(\hat{x}_\tau) + \sum_{\tau=1}^{p} \hat{c}_\tau(\hat{x}_\tau, \hat{x}_{\tau-1}, \hat{w}_{\tau-1}).
$$

To simplify the notation, we use $\hat{\zeta}$ to denote the tuple of system parameters, i.e.,

$$
\hat{\zeta} := (\hat{x}_0, \hat{w}_{0:p-1}, \hat{x}_p).
$$

From out construction, we know that $\hat{h}$ is $\mu$-strongly convex in $\hat{x}_{1:p-1}$, so we use the decomposition $\hat{h} = \hat{h}_a + \hat{h}_b$, where

$$
\hat{h}_a(\hat{x}_{1:p-1}, \hat{\zeta}) = \sum_{\tau=1}^{p-1} \frac{\mu}{2} \|\hat{x}_\tau\|^2 + \sum_{\tau=1}^{p} \hat{c}_\tau(\hat{x}_\tau, \hat{x}_{\tau-1}, \hat{w}_{\tau-1}),
$$

$$
\hat{h}_b(\hat{x}_{1:p-1}, \hat{\zeta}) = \sum_{\tau=1}^{p-1} \left( \hat{f}_\tau(\hat{x}_\tau) - \frac{\mu}{2} \|\hat{x}_\tau\|^2 \right).
$$

Since $\hat{\psi}(\hat{\zeta} + \theta e)$ is the minimizer of convex function $\hat{h}(\cdot, \hat{\zeta} + \theta e)$, we see that

$$
\nabla_{\hat{x}_{1:p-1}} \hat{h}(\hat{\psi}(\hat{\zeta} + \theta e), \hat{\zeta} + \theta e) = 0.
$$

Taking the derivative with respect to $\theta$ gives that

$$\nabla^2_{\hat{x}_{1:p-1}}\hat{h}(\hat{\psi}(\hat{\zeta}+\theta e),\hat{\zeta}+\theta e)\frac{d}{d\theta}\hat{\psi}(\hat{\zeta}+\theta e) = -\nabla_{\hat{x}_0}\nabla_{\hat{x}_{1:p-1}}\hat{h}(\hat{\psi}(\hat{\zeta}+\theta e),\hat{\zeta}+\theta e)e_0$$
$$-\nabla_{\hat{x}_p}\nabla_{\hat{x}_{1:p-1}}\hat{h}(\hat{\psi}(\hat{\zeta}+\theta e),\hat{\zeta}+\theta e)e_p$$
$$-\sum_{\tau=0}^{p-1}\nabla_{w_\tau}\nabla_{\hat{x}_{1:p-1}}\hat{h}(\hat{\psi}(\hat{\zeta}+\theta e),\hat{\zeta}+\theta e)\mu_\tau.$$

To simplify the notation, we define

$$M := \nabla^2_{\hat{x}_{1:p-1}}\hat{h}(\hat{\psi}(\hat{\zeta}+\theta e),\hat{\zeta}+\theta e), \text{ which is a } (p-1)\times(p-1) \text{ block matrix,}$$

$$R^{(0)} := -\nabla_{\hat{x}_0}\nabla_{\hat{x}_{1:p-1}}\hat{h}(\hat{\psi}(\hat{\zeta}+\theta e),\hat{\zeta}+\theta e), \text{ which is a } (p-1)\times 1 \text{ block matrix,}$$

$$R^{(p)} := -\nabla_{\hat{x}_p}\nabla_{\hat{x}_{1:p-1}}\hat{h}(\hat{\psi}(\hat{\zeta}+\theta e),\hat{\zeta}+\theta e), \text{ which is a } (p-1)\times 1 \text{ block matrix,}$$

$$K^{(\tau)} := -\nabla_{w_\tau}\nabla_{\hat{x}_{1:p-1}}\hat{h}(\hat{\psi}(\hat{\zeta}+\theta e),\hat{\zeta}+\theta e), \forall 0\le\tau\le p-1, \text{ which are } (p-1)\times 1 \text{ block matrices,}$$

where in $M$, $R^{(0)}$, $R^{(p)}$, the block size is $n\times n$; in $K^{(\tau)}$, the block size is $n\times r$. Hence we can write

$$\frac{d}{d\theta}\hat{\psi}(\hat{\zeta}+\theta e) = M^{-1}\left(R^{(0)}e_0 + R^{(p)}e_p + \sum_{\tau=0}^{p-1}K^{(\tau)}\mu_\tau\right).$$

Recall that $R^{(0)}$, $R^{(p)}$ are $(p-1)\times 1$ block matrices with block size $n\times n$. $\{K^{(\tau)}\}_{0\le\tau\le p-1}$ are $(p-1)\times 1$ block matrices with block size $n\times r$. For $R^{(0)}$ and $K^{(0)}$, only the $(1,1)$-th blocks are non-zero. For $R^{(p)}$ and $K^{(p-1)}$, only the $(p-1,1)$-th blocks are non-zero. For $K^{(\tau)}, \tau=1,\ldots,p-2$, only the $(\tau,1)$-th and $(\tau+1,1)$-th blocks are non-zero. Hence we see that

$$\frac{d}{d\theta}\hat{\psi}(\hat{\zeta}+\theta e)_h = (M^{-1})_{h,1}R^{(0)}_{1,1}e_0 + (M^{-1})_{h,p-1}R^{(p)}_{p-1,1}e_p$$
$$+ (M^{-1})_{h,1}K^{(0)}_{1,1}\mu_0 + (M^{-1})_{h,p-1}K^{(p-1)}_{p-1,1}\mu_{p-1}$$
$$+ \sum_{\tau=1}^{p-2}(M^{-1})_{h,\tau:\tau+1}K^{(\tau)}_{\tau:\tau+1,1}\mu_\tau.$$

Since the switching costs $c_\tau(\cdot,\cdot,\cdot),\tau=1,\ldots,p$ are $\ell$-strongly smooth, we know that the norms of

$$R^{(0)}_{1,1}, R^{(p)}_{p-1,1}, K^{(0)}_{1,1}, K^{(p-1)}_{p-1,1}, \text{ and } \{K^{(\tau)}_{\tau:\tau+1,1}\}_{1\le\tau\le p-2}$$

are all upper bounded by $\ell$. Taking norm on both sides gives that

$$\left\|\frac{d}{d\theta}\hat{\psi}(\hat{\zeta}+\theta e)_h\right\| \le \ell\left\|(M^{-1})_{h,1}\right\|\|e_0\| + \ell\left\|(M^{-1})_{h,p-1}\right\|\|e_p\|$$
$$+ \ell\left\|(M^{-1})_{h,1}\right\|\|\mu_0\| + \ell\left\|(M^{-1})_{h,p-1}\right\|\|\mu_{p-1}\|$$
$$+ \sum_{\tau=1}^{p-2}\ell\left\|(M^{-1})_{h,\tau:\tau+1}\right\|\|\mu_\tau\|. \qquad (8)$$

Note that $M$ can be decomposed as $M = M_a + M_b$, where

$$M_a := \nabla^2_{1:p-1}\hat{h}_a(\hat{\psi}(\hat{\zeta}+\theta e),\hat{\zeta}+\theta e),$$
$$M_b := \nabla^2_{1:p-1}\hat{h}_b(\hat{\psi}(\hat{\zeta}+\theta e),\hat{\zeta}+\theta e).$$

Since $M_a$ is block tri-diagonal and satisfies $(\mu+2\ell)I \ge M_a \ge \mu I$, and $M_b$ is block diagonal and satisfies $M_b \ge 0$, we obtain the following with Lemma B.1:

$$\left\|(M^{-1})_{h,1}\right\| \le \frac{2}{\mu}\lambda_0^{h-1}, \left\|(M^{-1})_{h,p-1}\right\| \le \frac{2}{\mu}\lambda_0^{p-h-1}, \text{ and } \left\|(M^{-1})_{h,\tau:\tau+1}\right\| \le \frac{2}{\mu}\lambda_0^{|h-\tau|-1},$$

where $\lambda_0 := (\sqrt{\text{cond}(M_a)}-1)/(\sqrt{\text{cond}(M_a)}+1) = 1 - 2\cdot\left(\sqrt{1+(2\ell/\mu)}+1\right)^{-1}$.

Substituting this into (8), we see that

$$\left\|\frac{d}{d\theta}\hat{\psi}(\hat{\zeta}+\theta e)_h\right\| \le C_0\left(\lambda_0^{h-1}\|e_0\| + \sum_{\tau=0}^{p-1}\lambda_0^{|h-\tau|-1}\|\mu_\tau\| + \lambda_0^{p-h-1}\|e_p\|\right),$$

where $C_0 = (2\ell)/\mu$.

Hence we obtain

$$\begin{aligned}
\left\|\hat{\psi}(\hat{\zeta})_h - \hat{\psi}(\hat{\zeta}+e)_h\right\| &= \left\|\int_0^1 \frac{d}{d\theta}\hat{\psi}(\hat{\zeta}+\theta e)_h\, d\theta\right\| \\
&\le \int_0^1 \left\|\frac{d}{d\theta}\hat{\psi}(\hat{\zeta}+\theta e)_h\right\| d\theta \\
&\le C_0\left(\lambda_0^{h-1}\|e_0\| + \sum_{\tau=0}^{p-1}\lambda_0^{|h-\tau|-1}\|\mu_\tau\| + \lambda_0^{p-h-1}\|e_p\|\right).
\end{aligned}$$

This finishes the proof. $\qquad\square$

## C   Proof of Lemma 3.2

**Lemma C.1.** *Suppose function $f(x,y)$ is convex and $L$-strongly smooth in $(x,y)$, $\mu$-strongly convex in $y$, and continuously differentiable. Define functions $y^*(x) := \arg\min_y f(x,y)$ and $g(x) := \min_y f(x,y)$. Then, function $y^*$ is $\frac{L}{\mu}$-Lipschitz and function $g$ is $\left(L + \frac{L^2}{\mu}\right)$-strongly smooth.*

*Proof of Lemma C.1.* Let $y^*(x) = \arg\min_y f(x,y)$. This function is well-defined since the strong convexity of $f(x,y)$ in $y$ guarantees that $y^*(x)$ is unique. We see that for all $x, x'$,

$$\nabla_y f(x, y^*(x)) = 0 \text{ and } \nabla_y f(x', y^*(x')) = 0.$$

Using these equalities, we obtain

$$\begin{aligned}
0 &= \langle y^*(x) - y^*(x'), \nabla_y f(x, y^*(x)) - \nabla_y f(x', y^*(x'))\rangle \\
&= \langle y^*(x) - y^*(x'), \nabla_y f(x, y^*(x)) - \nabla_y f(x, y^*(x'))\rangle \\
&\quad + \langle y^*(x) - y^*(x'), \nabla_y f(x, y^*(x')) - \nabla_y f(x', y^*(x'))\rangle \\
&\ge \mu\|y^*(x) - y^*(x')\|^2 - \|y^*(x) - y^*(x')\| \cdot \left\|\nabla_y f(x, y^*(x')) - \nabla_y f(x', y^*(x'))\right\|,
\end{aligned}$$

where we used the fact that a $\mu$-strongly convex function $h$ satisfies

$$\langle a - b, \nabla h(a) - \nabla h(b)\rangle \ge \mu\|a - b\|^2, \forall a, b$$

and the Cauchy-Schwartz inequality in the last inequality. Since $f$ is $L$-strongly smooth, we see that

$$\|y^*(x) - y^*(x')\| \le \frac{1}{\mu}\left\|\nabla_y f(x, y^*(x')) - \nabla_y f(x', y^*(x'))\right\| \le \frac{L}{\mu}\|x - x'\|,$$

which implies function $y^*$ is $\frac{L}{\mu}$-Lipschitz.

Note that the gradient of $g$ is given by

$$\nabla g(x) = \nabla_x f(x, y^*(x)) + \nabla_y f(x, y^*(x))\frac{\partial y^*(x)}{\partial x} = \nabla_x f(x, y^*(x)),$$

because $\nabla_y f(x, y^*(x)) = 0$. Hence we obtain

$$\begin{aligned}
\|\nabla g(x) - \nabla g(x')\| &\le \|\nabla_x f(x, y^*(x)) - \nabla_x f(x', y^*(x'))\| \\
&\le \|\nabla_x f(x, y^*(x)) - \nabla_x f(x', y^*(x))\| + \|\nabla_x f(x', y^*(x)) - \nabla_x f(x', y^*(x'))\| \\
&\le L\|x - x'\| + L\|y^*(x) - y^*(x')\| \\
&\le \left(L + \frac{L^2}{\mu}\right)\|x - x'\|.
\end{aligned}$$

$\qquad\square$

**Lemma C.2.** *Suppose $A$ is a $\omega_1 \times \omega_2$ block matrix. Let $A_{ij}$ denote the $(i, j)$ th block of $A$, $1 \leq i \leq \omega_1, 1 \leq j \leq \omega_2$. The induced 2-norm of $A$ is upper bounded by*

$$\|A\| \leq \left( \sum_{i=1}^{\omega_1} \sum_{j=1}^{\omega_2} \|A_{ij}\|^2 \right)^{\frac{1}{2}}.$$

*Proof of Lemma C.2.* For unit vector $x$, we have the following:

$$
\begin{aligned}
\|Ax\|^2 &= \sum_{i=1}^{\omega_1} \left\| \sum_{j=1}^{\omega_2} A_{ij} x_j \right\|^2 \\
&\leq \sum_{i=1}^{\omega_1} \left( \sum_{j=1}^{\omega_2} \|A_{ij}\| \cdot \|x_j\| \right)^2 \\
&\leq \sum_{i=1}^{\omega_1} \left( \sum_{j=1}^{\omega_2} \|A_{ij}\|^2 \right) \left( \sum_{j=1}^{\omega_2} \|x_j\|^2 \right) \\
&= \sum_{i=1}^{\omega_1} \sum_{j=1}^{\omega_2} \|A_{ij}\|^2.
\end{aligned}
$$

where we used the definition of the induced 2-norm in the first inequality and the Cauchy-Schwarz inequality in the second inequality. □

Now we come back to the proof of Lemma 3.2.

*Proof of Lemma 3.2.* To simplify the notation, we define the stacked state vector $y$, control vector $v$, and disturbance vector $\zeta$ as

$$
y = \begin{bmatrix} y_0 \\ y_1 \\ \vdots \\ y_p \end{bmatrix}, v = \begin{bmatrix} v_0 \\ v_1 \\ \vdots \\ v_{p-1} \end{bmatrix}, \zeta = \begin{bmatrix} \zeta_0 \\ \zeta_1 \\ \vdots \\ \zeta_{p-1} \end{bmatrix}.
$$

Recall that the transition matrix $\Phi(t_2, t_1)$ is defined as

$$
\Phi(t_2, t_1) := \begin{cases} A_{t_2-1} A_{t_2-2} \cdots A_{t_1} & \text{if } t_2 > t_1 \\ I & \text{if } t_2 \leq t_1 \end{cases}.
$$

Using this, we can express the state vector $y$ as an affine function of initial state $x$, control $v$, and disturbance $\zeta$:

$$y = S^x x + S^v v + S^\zeta \zeta, \tag{9}$$

where

$$
S^\zeta := \begin{bmatrix}
0 & 0 & \cdots & 0 \\
\Phi(t+1, t+1) & 0 & \cdots & 0 \\
\Phi(t+2, t+1) & \Phi(t+2, t+2) & \cdots & 0 \\
\vdots & \vdots & \ddots & \vdots \\
\Phi(t+p, t+1) & \Phi(t+p, t+2) & \cdots & \Phi(t+p, t+p)
\end{bmatrix}, S^x = \begin{bmatrix}
\Phi(t, t) \\
\Phi(t+1, t) \\
\Phi(t+2, t) \\
\vdots \\
\Phi(t+p, t)
\end{bmatrix},
$$

and $S^v = S^\zeta \cdot diag(B_t, \ldots, B_{t+p-1})$.

To simplify the notation, we use the shorthand $M := M(t, p)$ for the controllability matrix and

$$R^\zeta := [\Phi(t+p, t+1), \Phi(t+p, t+2), \ldots, \Phi(t+p, t+p)].$$

throughout the proof. Since $p$ is greater than the controllability index $d$, we know $M$ has full row rank. The dynamical constraints for (5), which is identical to the constraints of (3), can be written as

$$Mv = z - \Phi(t+p, t)x - R^\zeta \zeta.$$

Because $M$ has full row rank, we let $M^\dagger = M^\top (MM^\top)^{-1}$ be the Moore-Penrose pseudo-inverse of $M$. Let $V \in \mathbb{R}^{(mp) \times (mp-n)}$ be a matrix whose columns constitute an orthonormal basis of $ker(M)$. Then, we can express all feasible control vector $v$ as

$$v = M^\dagger \left( z - \Phi(t+p, t)x - R^\zeta \zeta \right) + Vr, \tag{10}$$

where $r$ is a free variable that can take any value in $\mathbb{R}^{mp-n}$.

Let $F$ denote the objective function of $\xi_t^p$, i.e.,

$$F(y, v) := \left( \sum_{\tau=1}^{p-1} f_{t+\tau}(y_\tau) + c_{t+\tau}(v_{\tau-1}) \right) + c_{t+p}(v_{p-1}).$$

Since we can express the state vector $y$ and control vector $v$ as linear functions of $x, z, \zeta$ and $r$, we can write the switching cost (5) as an unconstrained optimization problem

$$\min_{r \in \mathbb{R}^{mp-n}} F(y(x, z, \zeta, r), v(x, z, \zeta, r)), \tag{11}$$

where functions $y(x, z, \zeta, r)$ and $v(x, z, \zeta, r)$ are determined by

$$\begin{bmatrix} y \\ v \end{bmatrix} = \begin{bmatrix} S^x - S^v M^\dagger \Phi(t+p, t) & S^v M^\dagger & S^\zeta - S^v M^\dagger R^\zeta & S^v V \\ -M^\dagger \Phi(t+p, t) & M^\dagger & -M^\dagger R^\zeta & V \end{bmatrix} \cdot \begin{bmatrix} x \\ z \\ \zeta \\ r \end{bmatrix}. \tag{12}$$

Note that if $a \neq 1$, the following is due to Lemma C.2:

$$\|S^\zeta\| \leq \left( \sum_{i=1}^{p} \sum_{j=1}^{i} \|\phi(t+i, t+j)\|^2 \right)^{\frac{1}{2}} \leq \left( \sum_{i=1}^{p} \sum_{j=1}^{i} a^{2(i-j)} \right)^{\frac{1}{2}} = \frac{\sqrt{a^{2p+2} - (p+1)a^2 + p}}{|a^2 - 1|}$$

By Lemma C.2, we also have

$$\|S^x\| \leq \sqrt{\frac{a^{2p+2} - 1}{a^2 - 1}}, \|M^\dagger\| \leq \frac{b}{\sigma^2} \cdot \sqrt{\frac{a^{2p} - 1}{a^2 - 1}}, \|S^v\| \leq b\|S^\zeta\|, \|R^\zeta\| \leq \sqrt{\frac{a^{2p} - 1}{a^2 - 1}} \leq \frac{a^p - 1}{a - 1}.$$

Since the norm of a block matrix is upper bounded by the sum of norms of each block, we see that

$$\left\| \begin{bmatrix} S^x - S^v M^\dagger \Phi(t+p, t) & S^v M^\dagger & S^\zeta - S^v M^\dagger R^\zeta & S^v V \\ -M^\dagger \Phi(t+p, t) & M^\dagger & -M^\dagger R^\zeta & V \end{bmatrix} \right\| \leq C(p), \tag{13}$$

where, when $a \neq 1$,

$$C(p) = \left( \frac{b(a^{p+1} + a - 2)}{\sigma^2(a - 1)} \cdot \sqrt{\frac{a^{2p} - 1}{a^2 - 1}} + \frac{1 + b}{b} \right) \left( \frac{b\sqrt{(a^{2p+2} - (p+1)a^2 + p)}}{|a^2 - 1|} + 1 \right) + \sqrt{\frac{a^{2p+2} - 1}{a^2 - 1}} - \frac{1}{b}.$$

If $a = 1$, by Lemma C.2, we see that

$$\|S^\zeta\| \leq \left( \sum_{i=1}^{p} \sum_{j=1}^{i} \|\phi(t+i, t+j)\|^2 \right)^{\frac{1}{2}} \leq \left( \sum_{i=1}^{p} \sum_{j=1}^{i} a^{2(i-j)} \right)^{\frac{1}{2}} = \sqrt{\frac{p(p+1)}{2}}.$$

By Lemma C.2, we also see that

$$\|S^x\| \leq \sqrt{p+1}, \|M^\dagger\| \leq \frac{b}{\sigma^2} \cdot \sqrt{p}, \|S^v\| \leq b\|S^\zeta\|, \|R^\zeta\| \leq \sqrt{p}.$$

Therefore, for (13) to hold when $a = 1$, we need to set

$$C(p) = \left( \frac{b\sqrt{p}}{\sigma^2} (\sqrt{p} + 2) + 1 \right) \left( 1 + b\sqrt{\frac{p(p+1)}{2}} \right) + \sqrt{p+1} \cdot \left( 1 + \sqrt{\frac{p}{2}} \right).$$

Since $F$ is convex and strongly smooth in $(x, u)$, and both $x, u$ are affine functions of $(y, z, r)$, $F(x(y, z, r), u(y, z, r))$ is convex and $\ell \cdot C(p)^2$-strongly smooth in $(y, z, r)$. Since $F(x, u)$ is $m_c$-strongly convex in $u$, by (10), we have

$$\nabla_r^2 F(x(y, z, w, r), u(y, z, w, r)) \succeq V^\top \nabla_u^2 F(x, u) V$$
$$\succeq m_c I,$$

where we used that $\|Vv\|_2 = \|v\|_2, \forall v \in \mathbb{R}^{mp-n}$ because the columns of $V$ are orthonormal in the last inequality. Therefore, by Lemma C.1, we know that (11) is convex and $L_2(p)$-strongly smooth in $(y, z)$, where

$$L_2(p) := \ell \cdot C(p)^2 + \frac{\ell^2 \cdot C(p)^4}{m_c}.$$

By Lemma C.1, we also know that the optimal solution of (11):

$$r^*(x, z, \zeta) := \arg\min_{r \in \mathbb{R}^{mp-n}} F(y(x, z, \zeta, r), v(x, z, \zeta, r))$$

is $\ell \cdot C(p)^2/m_c$-Lipschitz. By (12) and (13), we see that

$$\psi_t^P(x, \zeta, z) = \begin{bmatrix} S^x - S^v M^\dagger \Phi(t + p, t) & S^v M^\dagger & S^\zeta - S^v M^\dagger R^\zeta & S^v V \\ -M^\dagger \Phi(t + p, t) & M^\dagger & -M^\dagger R^\zeta & V \end{bmatrix} \cdot \begin{bmatrix} x \\ z \\ \zeta \\ r^*(x, z, \zeta) \end{bmatrix}$$

is $L_1(p)$-Lipschitz, where

$$L_1(p) = C(p)(1 + \ell \cdot C(p)^2/m_c).$$

# D Proof of Theorem 3.3

The proof of Theorem 3.3 is based on the decision-point transformation introduced in Section 3.2.

Recall that we use $d$ to the controllability index as defined in Definition 2.1. To show the perturbation bound of $\psi_t^P(\cdot, \cdot, \cdot)_{y_h}$, suppose $h$ and $p$ satisfy $ud \leq h < (u + 1)d$ and $p = vd + r$, where $u, v, r \in \mathbb{N}$ and $0 \leq r < d$. Now we shall select the decision points as

$$y_0, y_d, \cdots, y_{(u-1)d}, y_h, y_{(u+2)d}, \cdots, y_{(v-1)d}, y_p,$$

which are also denoted by $y_{i_0}, \cdots, y_{i_{v-1}}$ for simplicity. Since the distance of any consecutive decision points falls in $[d, 2d]$, we can apply Lemma 3.2 to bound the strong smoothness of switching costs. In the transformed SOCO problem, the disturbance input of the $(\tau - 1)$-th time period is a vector $\hat{w}_{\tau-1} = \zeta_{i_{\tau-1}:i_\tau-1} \in \mathbb{R}^{n \times (i_\tau - i_{\tau-1})}$. Each stage cost $\xi_t^{i_\tau - i_{\tau-1}}(x_{i_{\tau-1}}, \hat{w}_{\tau-1}, x_{i_\tau})$ is convex and $L_2(i_\tau - i_{\tau-1})$-strongly smooth by Lemma 3.2, and is thus $L_0$-strongly smooth by definition. Recall that the solution of the transformed SOCO problem is denoted by $\hat{\psi}(x_t, \zeta, x_{t+p})$. Then by Theorem 3.1 we have

$$\left\| \psi_t^P(x, \zeta, z)_{y_h} - \psi_t^P(x', \zeta', z')_{y_h} \right\|$$
$$= \left\| \hat{\psi}(x, \zeta, z)_u - \hat{\psi}(x', \zeta', z')_u \right\|$$
$$\leq C_0 \left( \lambda_0^{u-1} \|x - x'\|_2 + \sum_{\tau=0}^{v-2} \lambda_0^{|u-\tau|-1} \|w_\tau - w_\tau'\|_2 + \lambda_0^{(v-1)-u-1} \|z - z'\|_2 \right)$$
$$= C_0 \left( \lambda_0^{u-1} \|x - x'\|_2 + \sum_{\tau=0}^{v-2} \lambda_0^{|u-\tau|-1} \sum_{j=i_\tau}^{i_{\tau+1}-1} \left\| \zeta_j - \zeta_j' \right\|_2 + \lambda_0^{(v-1)-u-1} \|z - z'\|_2 \right)$$
$$\leq \frac{C_0}{\lambda_0} \left( \lambda^{i_u-i_0} \|x - x'\|_2 + \sum_{\tau=0}^{v-2} \sum_{j=i_\tau}^{i_{\tau+1}-1} \lambda^{|j-i_u|} \left\| \zeta_j - \zeta_j' \right\|_2 + \lambda^{i_{v-1}-i_u} \|z - z'\|_2 \right)$$
$$= C \left( \lambda^h \|x - x'\| + \sum_{\tau=0}^{p-1} \lambda^{|h-\tau|} \left\| \zeta_\tau - \zeta_\tau' \right\| + \lambda^{p-h} \|z - z'\| \right).$$

The last inequality holds because each interval is of length at most $(2d - 1)$. Here the constants are

$$C_0 = \frac{2L_0}{m_c}, \lambda_0 = 1 - 2 \cdot \left( \sqrt{1 + (2L_0/m_c)} + 1 \right)^{-1},$$

$$C = C_0/\lambda_0 = \frac{2L_0}{m_c}\left(1 - 2 \cdot \left(\sqrt{1 + (2L_0/m_c)} + 1\right)^{-1}\right)^{-1}, \lambda = \left(1 - 2\left(\sqrt{1 + (2L_0/m_c)} + 1\right)^{-1}\right)^{\frac{1}{2d-1}}.$$

The proof of the perturbation bound of $\psi_t^p(\cdot,\cdot,\cdot)_{y_h}$ is quite similar. The only difference lies in the terminal cost, which can be addressed with the addition of a fixed auxiliary state. Specifically, we append $x_{\text{aux}} = 0$ to the end of the decision point sequence, and define a zero transition cost to the auxiliary state $\hat{c}_v(x_{t+p}, w_{v-1}, x_{\text{aux}}) \equiv 0$ (note that $\hat{c}_v$ is trivially convex and $L_0$-strongly smooth). Denote the solution of the modified version of transformed SOCO problem by $\hat{\psi}'(x_t, \zeta, x_{\text{aux}})$, then by the same argument as above, we have

$$\left\|\tilde{\psi}_t^P(x,\zeta)_{y_h} - \tilde{\psi}_t^P(x',\zeta')_{y_h}\right\| = \left\|\hat{\psi}'(x,\zeta,0)_u - \hat{\psi}'(x',\zeta',0)_u\right\|$$

$$\leq \cdots \leq C\left(\lambda^h\|x - x'\| + \sum_{\tau=0}^{p-1} \lambda^{|h-\tau|}\|\zeta_\tau - \zeta_\tau'\|\right),$$

where the constants are the same as previously defined. This finishes the proof of Theorem 3.3.

# E    Stability of the Optimal Trajectory

**Corollary E.1** (Stability of the Optimal Trajectory). *For the predicted trajectory from solving* (2) *with a prediction window $p \geq d$, the norm of the h-th predictive state is bounded above by*

$$\left\|\tilde{\psi}_t^P(x,\zeta;F)_{y_h}\right\| \leq C\left(\lambda^h\|x\| + \sum_{\tau=0}^{p-1} \lambda^{|h-\tau|}\|\zeta_\tau\|\right) \leq C\lambda^h\|x\| + \frac{2C}{1-\lambda}\sup_\tau\|\zeta_\tau\|.$$

*Proof of Corollary E.1.* Note that $\tilde{\psi}_t^P(0,0)_{y_h} = 0$. By Theorem 3.3, we see that

$$\left\|\tilde{\psi}_t^P(x,\zeta)_{y_h}\right\| = \left\|\tilde{\psi}_t^P(x,\zeta)_{y_h} - \tilde{\psi}_t^P(0,0)_{y_h}\right\|$$

$$\leq C\left(\lambda_1^h\|x\| + \sum_{\tau=0}^{p-1} \lambda_1^{|h-\tau|}\|\zeta_\tau\|\right)$$

$$\leq C\lambda_1^h\|x\| + \frac{2C}{1-\lambda_1}\sup_\tau\|\zeta_\tau\|,$$

where the last inequality holds because

$$\sum_{\tau=0}^{p-1} \lambda_1^{|h-\tau|} \leq \frac{2}{1-\lambda_1}.$$

$\square$

# F    Smoothness of the Optimal Cost

**Corollary F.1.** *For any time step t and integer p that satisfies $p \geq d$, function $\iota_t^p(\cdot,\zeta,\cdot)$ satisfies that*

$$\iota_t^p(x,\zeta,z) \leq (1+\eta)\iota_t^p(x',\zeta,z') + \frac{L_0 + \ell_f}{2}\left(1 + \frac{1}{\eta}\right)\left(\|x' - x\|^2 + \|z' - z\|^2\right), \forall x, x', \zeta, z, z'.$$

**Lemma F.2.** *Assume a function $g : \mathbb{R}^n \to \mathbb{R}_+$ is convex, $\ell$-strongly smooth and continuously differentiable. For all $x, y \in \mathbb{R}^n$, for all $\eta > 0$, we have*

$$g(x) \leq (1+\eta)g(y) + \frac{\ell}{2}\left(1 + \frac{1}{\eta}\right)\|x - y\|^2.$$

*Proof of Lemma F.2.*

$$g(x) - g(y) \leq \langle \nabla g(y), x - y \rangle + \frac{\ell}{2}\|x - y\|^2$$

$$\leq \frac{\eta}{2\ell}\|\nabla g(y)\|^2 + \frac{\ell}{2\eta}\|x-y\|^2 + \frac{\ell}{2}\|x-y\|^2$$

$$\leq \eta g(y) + \frac{\ell}{2}\left(1+\frac{1}{\eta}\right)\|x-y\|^2.$$

where the second inequality follows from the generalized means inequality and the last inequality holds because

$$0 \leq g\left(y-\frac{1}{\ell}\nabla g(y)\right) \leq g(y) - \left\langle \nabla g(y), \frac{1}{\ell}\nabla g(y)\right\rangle + \frac{\ell}{2}\left\|\frac{\nabla g(y)}{\ell}\right\|^2 = g(y) - \frac{1}{2\ell}\|\nabla g(y)\|^2$$

$\square$

Now we come back to the proof of Corollary F.1.

When $d \leq p \leq 2d-1$, since $\xi_t^P(x,\zeta,z)$ is $L_0$-strongly smooth by Lemma 3.2, we know

$$\iota_t^P(x,\zeta,z) = \xi_t^P(x,\zeta,z) + f_{t+p}(z)$$

is $(L_0 + \ell_f)$-strongly smooth. Therefore, by Lemma F.2, we obtain that

$$\iota_t^P(x,\zeta,z) \leq (1+\eta)\iota_t^P(x',\zeta,z') + \frac{L_0+\ell_f}{2}\left(1+\frac{1}{\eta}\right)\left(\|x'-x\|^2 + \|z'-z\|^2\right).$$

When $p = 2d$, let $x_1 := \psi_t^P(x',\zeta,z')_{y_d}$. We see that

$$\iota_t^P(x,\zeta,z) \leq \iota_t^d(x,\zeta_{0:d-1},x_1) + \iota_{t+d}^d(x_1,\zeta_{d:2d-1},z)$$

$$\leq (1+\eta)\iota_t^d(x',\zeta_{0:d-1},x_1) + \frac{L_0+\ell_f}{2}\left(1+\frac{1}{\eta}\right)\|x-x'\|^2$$

$$+ (1+\eta)\iota_{t+d}^d(x_1,\zeta_{d:2d-1},z') + \frac{L_0+\ell_f}{2}\left(1+\frac{1}{\eta}\right)\|z-z'\|^2$$

$$\leq (1+\eta)\iota_t^P(x',\zeta,z') + \frac{L_0+\ell_f}{2}\left(1+\frac{1}{\eta}\right)\left(\|x'-x\|^2 + \|z'-z\|^2\right).$$

When $p > 2d$, let $x_1 := \psi_t^P(x',\zeta,z')_{y_d}, x_2 := \psi_t^P(x',\zeta,z')_{y_{p-d}}$. We see that

$$\iota_t^P(x,\zeta,z) \leq \iota_t^d(x,\zeta_{0:d-1},x_1) + \iota_{t+d}^{p-2d}(x_1,\zeta_{d:p-d-1},x_2) + \iota_{t+p-d}^d(x_2,\zeta_{p-d:p-1},z)$$

$$\leq (1+\eta)\iota_t^d(x',\zeta_{0:d-1},x_1) + \frac{L_0+\ell_f}{2}\left(1+\frac{1}{\eta}\right)\|x-x'\|^2$$

$$+ \iota_{t+d}^{p-2d}(x_1,\zeta_{d:p-d-1},x_2)$$

$$+ (1+\eta)\iota_{t+p-d}^d(x_2,\zeta_{p-d:p-1},z') + \frac{L_0+\ell_f}{2}\left(1+\frac{1}{\eta}\right)\|z-z'\|^2$$

$$\leq (1+\eta)\iota_t^P(x',\zeta,z') + \frac{L_0+\ell_f}{2}\left(1+\frac{1}{\eta}\right)\left(\|x'-x\|^2 + \|z'-z\|^2\right).$$

# G   Proof of Lemma 4.1

*Proof of Lemma 4.1.* To simplify the notation, we define

$$z := \tilde{\psi}_t^P(x_t;F)_{y_p}, z' := \tilde{\psi}_t^{P+1}(x_t;F)_{y_p}.$$

We see that

$$\left\|\tilde{\psi}_t^P(x_t;F)_{y_h} - \tilde{\psi}_t^{P+1}(x_t;F)_{y_h}\right\| = \left\|\psi_t^P(x_t,z)_{y_h} - \psi_t^P(x_t,z')_{y_h}\right\| \tag{14a}$$

$$\leq C\lambda^{p-h}\|z-z'\| \tag{14b}$$

$$\leq 2C\lambda^{p-h}\left(C\lambda^p\|x_t\| + \frac{2C}{1-\lambda}D\right). \tag{14c}$$

where we used the definition of $\psi$ and $\tilde{\psi}$ in (14a); Theorem 3.3 in (14b); Corollary E.1 in (14c). $\square$

# H Proof of Theorem 4.2

Throughout the proof, we will use $\{(\hat{x}_t, \hat{u}_t)\}$ to denote the trajectory of predictive control with prediction window $T$ ($PC_T$). Recall that $\{(x_t, u_t)\}$ denotes the trajectory of predictive control with prediction window $k$ ($PC_k$), and $\{(x_t^*, u_t^*)\}$ denotes the offline optimal trajectory ($OPT$), i.e., the optimal solution of (1).

For simplicity, we will use the shorthand notations

$$\tilde{\psi}_t^p(x; F) := \tilde{\psi}_t^p(x, w_{t:t+p-1}; F) \text{ and } \psi_t^p(x, z) := \psi_t^p(x, w_{t:t+p-1}, z)$$

throughout the proof.

*Proof of Theorem 4.2.* Since $x_{t+1} = \tilde{\psi}_t^p(x_t; F)_{y_1}$, for all $2 \le i \le k$, we have

$$\tilde{\psi}_{t-i}^k(x_{t-i}; F)_{y_i} = \tilde{\psi}_{t-i+1}^{k-1}(x_{t-i+1}; F)_{y_{i-1}}. \tag{15}$$

Therefore, we obtain that for $k \le t \le T - k$,

$$\|x_t\| = \left\| \tilde{\psi}_{t-1}^k(x_{t-1}; F)_{y_1} \right\|$$

$$\le \sum_{i=1}^{k-1} \left\| \tilde{\psi}_{t-i}^k(x_{t-i}; F)_{y_i} - \tilde{\psi}_{t-i-1}^k(x_{t-i-1}; F)_{y_{i+1}} \right\| + \left\| \tilde{\psi}_{t-k}^k(x_{t-k}; F)_{y_k} \right\| \tag{16a}$$

$$= \sum_{i=1}^{k-1} \left\| \tilde{\psi}_{t-i}^k(x_{t-i}; F)_{y_i} - \tilde{\psi}_{t-i}^{k-1}(x_{t-i}; F)_{y_i} \right\| + \left\| \tilde{\psi}_{t-k}^k(x_{t-k}; F)_{y_k} \right\| \tag{16b}$$

$$\le \sum_{i=1}^{k-1} 2C\lambda^{k-1-i} \left( C\lambda^{k-1} \|x_{t-i}\| + \frac{2C}{1-\lambda} D \right) + \left( C\lambda^k \|x_{t-k}\| + \frac{2C}{1-\lambda} D \right) \tag{16c}$$

$$\le C\lambda^{k-1} \left( \lambda \|x_{t-k}\| + 2C \sum_{i=1}^{k-1} \lambda^{k-1-i} \|x_{t-i}\| \right) + \frac{2C}{1-\lambda} \left( 1 + \frac{2C}{1-\lambda} \right) D,$$

where we used the triangle inequality in (16a); (15) in (16b); Lemma 4.1 and Corollary E.1 in (16c). By a similar argument, for $1 \le t \le k$, we have

$$\|x_t\| = \left\| \tilde{\psi}_{t-1}^k(x_{t-1}; F)_{y_1} \right\|$$

$$\le \sum_{i=1}^{t-1} \left\| \tilde{\psi}_{t-i}^k(x_{t-i}; F)_{y_i} - \tilde{\psi}_{t-i-1}^k(x_{t-i-1}; F)_{y_{i+1}} \right\| + \left\| \tilde{\psi}_0^k(x_0; F)_{y_t} \right\|$$

$$= \sum_{i=1}^{t-1} \left\| \tilde{\psi}_{t-i}^k(x_{t-i}; F)_{y_i} - \tilde{\psi}_{t-i}^{k-1}(x_{t-i}; F)_{y_i} \right\| + \left\| \tilde{\psi}_0^k(x_0; F)_{y_t} \right\|$$

$$\le \sum_{i=1}^{t-1} 2C\lambda^{k-1-i} \left( C\lambda^{k-1} \|x_{t-i}\| + \frac{2C}{1-\lambda} D \right) + \left( C\lambda^t \|x_0\| + \frac{2C}{1-\lambda} D \right)$$

$$\le C\lambda^{k-1} \left( 2C \sum_{i=1}^{t-1} \lambda^{k-1-i} \|x_{t-i}\| \right) + C \|x_0\| + \frac{2C}{1-\lambda} \left( 1 + \frac{2C}{1-\lambda} \right) D, \tag{17}$$

Recall that, under the assumption of (6), the sum of coefficients in (16) and (17) are upper bounded by

$$C\lambda^{k-1} \left( 2C \sum_{i=1}^{t-1} \lambda^{k-1-i} \right) \le 1 - \delta, \quad C\lambda^{k-1} \left( \lambda + 2C \sum_{i=1}^{k-1} \lambda^{k-1-i} \right) < 1 - \delta.$$

Using inequalities (16) and (17), one can show by induction that for $t \le T - k$

$$\|x_t\| \le \frac{C}{\delta} \cdot (1-\delta)^{\max(0, t-k)} \|x_0\| + \frac{2C}{\delta(1-\lambda)} \left( 1 + \frac{2C}{1-\lambda} \right) D. \tag{18}$$

For $t \geq T - k + 1$, by Corollary E.1, we see that

$$
\begin{aligned}
\|x_t\| &= \left\| \tilde{\psi}_{T-k}^k (x_{T-k}, w_{T-k:T-1}, 0)_{y_{t+k-T}} \right\| \\
&\leq C\lambda^{t+k-T} \|x_{T-k}\| + \frac{2CD}{1-\lambda} \\
&\leq \frac{C^2}{\delta} \cdot (1-\delta)^{T-2k} \lambda^{t+k-T} \|x_0\| + \left( \frac{2C^2}{\delta(1-\lambda)} \left( 1 + \frac{2C}{1-\lambda} \right) + \frac{2C}{1-\lambda} \right) D.
\end{aligned}
$$

This finishes the proof of ISS of $PC_k$.

By Lemma 4.1 and (18), we also see that for $t \leq T - k$,

$$
\begin{aligned}
\left\| \tilde{\psi}_t^k(x_t; F)_{y_1} - \tilde{\psi}_t^{T-t}(x_t; F)_{y_1} \right\| &\leq \sum_{p=k}^{T-t} \left\| \tilde{\psi}_t^p(x_t; F)_{y_1} - \tilde{\psi}_t^{p+1}(x_t; F)_{y_1} \right\| \\
&\leq \sum_{p=k}^{\infty} 2C\lambda^{p-1} \left( C\lambda^p \|x_t\| + \frac{2C}{1-\lambda} D \right) \\
&= \frac{2C^2}{\lambda(1-\lambda^2)} \cdot (\lambda)^{2k} \|x_t\| + \frac{4C^2}{\lambda(1-\lambda)^2} \cdot \lambda^k D \\
&= O\left( \left( D + \frac{\lambda^k(\|x_0\| + D)}{\delta} \right) \lambda^k \right). \tag{19}
\end{aligned}
$$

We further obtain that for $t \leq T - k$,

$$
\begin{aligned}
\|x_t - \hat{x}_t\| &= \left\| x_t - \tilde{\psi}_0^T(x_0; F) \right\| \\
&\leq \left\| x_t - \tilde{\psi}_{t-1}^{T-t+1}(x_{t-1}; F)_{y_1} \right\| + \sum_{i=1}^{t-1} \left\| \tilde{\psi}_{t-i}^{T-t+i}(x_{t-i}; F)_{y_i} - \tilde{\psi}_{t-i-1}^{T-t+i+1}(x_{t-i-1}; F)_{y_{i+1}} \right\| \\
&\leq \left\| x_t - \tilde{\psi}_{t-1}^{T-t+1}(x_{t-1}; F)_{y_1} \right\| + \sum_{i=1}^{t-1} C\lambda^i \left\| x_{t-i} - \tilde{\psi}_{t-i-1}^{T-t+i+1}(x_{t-i-1}; F)_{y_1} \right\| \tag{20a} \\
&= O\left( \left( D + \frac{\lambda^k(\|x_0\| + D)}{\delta} \right) \lambda^k \right), \tag{20b}
\end{aligned}
$$

where we used Theorem 3.3 and the fact that $\tilde{\psi}_{t-i-1}^{T-t+i+1}(x_{t-i-1})_{y_{i+1}}$ can be written as

$$
\tilde{\psi}_{t-i-1}^{T-t+i+1}(x_{t-i-1}; F)_{y_{i+1}} = \tilde{\psi}_{t-i}^{T-t+i} \left( \tilde{\psi}_{t-i-1}^{T-t+i+1}(x_{t-i-1}; F)_{y_1}; F \right)_{y_i}
$$

in (20a); we used we used (19),

$$
\left\| x_{t-i} - \tilde{\psi}_{t-i-1}^{T-t+i+1}(x_{t-i-1}; F)_{y_1} \right\| = \left\| \tilde{\psi}_{t-i-1}^k(x_{t-i-1}; F)_{y_1} - \tilde{\psi}_{t-i-1}^{T-t+i+1}(x_{t-i-1}; F)_{y_1} \right\|
$$

and

$$
1 + \sum_{i=1}^{t-1} C\lambda^i \leq 1 + \frac{C}{1-\lambda} = O(1)
$$

in (20b).

By Corollary E.1, we see that

$$
\left\| x_T^* - \hat{x}_T \right\| \leq 2C\lambda^T \|x_0\| + \frac{4CD}{1-\lambda}.
$$

It follows that, by Theorem 3.3, the following holds for all $t \leq T - k$:

$$
\left\| x_t^* - \hat{x}_t \right\| = \left\| \psi_0^T(x_0, x_T^*) - \psi_0^T(x_0, \hat{x}_T) \right\| \leq C\lambda^k \left( 2C\lambda^T \|x_0\| + \frac{4CD}{1-\lambda} \right).
$$

Combining this inequality with (20) gives

$$
\|x_t - x_t^*\| = O\left( \left( D + \frac{\lambda^k(\|x_0\| + D)}{\delta} \right) \lambda^k \right), \quad \forall t \leq T - k. \tag{21}
$$

Since
$$(u_t - u_t^*) = B_t^\dagger\big((x_{t+1} - x_{t+1}^*) - A_t(x_t - x_t^*)\big),$$
we have
$$\|u_t - u_t^*\| \le b'\big(\|x_{t+1} - x_{t+1}^*\| + a\|x_t - x_t^*\|\big).$$
Therefore, under Corollary F.1, for any $\eta > 0$,

$$
\begin{aligned}
&\iota_t^1(x_t, x_{t+1}) - (1+\eta)\iota_t^1(x_t^*, x_{t+1}^*) \tag{22}\\
&= \big(f_{t+1}(x_{t+1}) - (1+\eta)f_{t+1}(x_{t+1}^*)\big) + \big(c_{t+1}(u_t) - (1+\eta)c_{t+1}(u_t^*)\big)\\
&\le \frac{1}{2}\Big(1 + \frac{1}{\eta}\Big)\Big(\ell_f\|x_{t+1} - x_{t+1}^*\|^2 + \ell_c\|u_t - u_t^*\|^2\Big)\\
&\le \frac{1}{2}\Big(1 + \frac{1}{\eta}\Big)\big(\ell_f + 2(b')^2\ell_c\big)\|x_{t+1} - x_{t+1}^*\|^2 + \frac{1}{2}\Big(1 + \frac{1}{\eta}\Big)2a^2(b')^2\ell_c\|x_t - x_t^*\|^2\\
&\le \Big(1 + \frac{1}{\eta}\Big)\cdot\frac{L_4}{2}\big(\|x_t - x_t^*\|^2 + \|x_{t+1} - x_{t+1}^*\|^2\big),
\end{aligned}
$$

where
$$L_4 := \ell_f + 2(b')^2\ell_c + 2a^2(b')^2\ell_c.$$

Then, for any $\eta > 0$, we obtain the following inequality:

$$
\begin{aligned}
&\text{cost}(PC_k) - (1+\eta)\,\text{cost}(OPT)\\
&= \left(\sum_{t=0}^{T-k-1}\iota_t^1(x_t, x_{t+1}) + \iota_{T-k}^k(x_{T-k}, x_T)\right) - (1+\eta)\left(\sum_{t=0}^{T-k-1}\iota_t^1(x_t^*, x_{t+1}^*) + \iota_{T-k}^k(x_{T-k}^*, x_T^*)\right)\\
&= \sum_{t=0}^{T-k-1}\Big(\iota_t^1(x_t, x_{t+1}) - (1+\eta)\iota_t^1(x_t^*, x_{t+1}^*)\Big) + \Big(\iota_{T-k}^k(x_{T-k}, x_T) - (1+\eta)\iota_{T-k}^k(x_{T-k}^*, x_T^*)\Big)\\
&\le \sum_{t=0}^{T-k-1}\Big(\iota_t^1(x_t, x_{t+1}) - (1+\eta)\iota_t^1(x_t^*, x_{t+1}^*)\Big) + \Big(\iota_{T-k}^k(x_{T-k}, x_T^*) - (1+\eta)\iota_{T-k}^k(x_{T-k}^*, x_T^*)\Big) \tag{23a}\\
&\le \Big(1 + \frac{1}{\eta}\Big)\cdot\frac{L_4}{2}\sum_{t=0}^{T-k-1}\big(\|x_t - x_t^*\|^2 + \|x_{t+1} - x_{t+1}^*\|^2\big) + \Big(1 + \frac{1}{\eta}\Big)\cdot\frac{L_0 + \ell_f}{2}\|x_{T-k} - x_{T-k}^*\|^2 \tag{23b}\\
&= \Big(1 + \frac{1}{\eta}\Big)\cdot L_4\sum_{t=0}^{T-k-1}\|x_t - x_t^*\|^2 + \Big(1 + \frac{1}{\eta}\Big)\cdot\frac{L_0 + \ell_f}{2}\|x_{T-k} - x_{T-k}^*\|^2\\
&\le \Big(1 + \frac{1}{\eta}\Big)O\!\left(\Big(D + \frac{\lambda^k(\|x_0\| + D)}{\delta}\Big)^2\lambda^{2k}T\right), \tag{23c}
\end{aligned}
$$

where we used the fact that the cost of $\tilde\psi_{T-k}^k(x_{T-k}; 0)$ is less than or equal to the cost of $\psi_{T-k}^k(x_{T-k}, x_T^*)$ in (23a); we used (22) and Corollary F.1 in (23b); and we used (20) in (23c). $\qquad\square$

To bound the optimal cost, we consider a suboptimal controller inspired by the decision-point transformation, where the controller forces the states $x_d, x_{2d}, \cdots, x_{(v-1)d}, x_{vd+r}$ to be 0 ($d$ is the controllability index, and $T = vd + r$). The cost of this suboptimal control is determined by the transformed transition cost $\xi_t^p(\cdot, \cdot, \cdot)$ between each pair of consecutive decision points. By strong smoothness of $\xi_t^p(\cdot, \cdot, \cdot)$ proven in Lemma 3.2, we have

$$\xi_t^p(x, \zeta, 0) \le \frac{1}{2}L_2(p)\big(\|\zeta\|^2 + \|x\|^2\big) \le \frac{L_0 D^2}{2}p + \frac{L_0}{2}\|x\|^2,$$

where $L_0 = \max_{d \le p \le 2d-1} L_2(p)$. These inequalities add up to

$$\text{cost}(OPT) \le \xi_0^d(x_0, w_{0:d-1}, 0) + \sum_{\tau=1}^{v-2}\xi_{\tau d}^d(0, w_{\tau d:(\tau+1)d-1}, 0) + \xi_{(t-1)d}^{d+r}(0, w_{(v-1)d:T-1}, 0)$$

$$\leq \frac{L_0 D^2}{2} T + \frac{L_0}{2} \|x_0\|^2$$
$$= O(D^2 T + \|x_0\|^2).$$

Hence $\text{cost}(OPT) = O(D^2 T + \|x_0\|^2)$. Now we can take $\eta = \Theta(\lambda^k)$ in (23) to get a regret bound of

$$\text{cost}(PC_k) - \text{cost}(OPT) = O\left(\left(D + \frac{\lambda^k(\|x_0\| + D)}{\delta}\right)^2 \lambda^k T + \lambda^k \|x_0\|^2\right).$$

# I  Proof of Theorem 4.3

*Proof of Theorem 4.3.* To simplify the notation, we will omit the disturbance sequence $w_{t:t+k-1}$ in $\tilde{\psi}_t^k$ and $\psi_t^k$ throughout the proof. At each time step $t$, we will use $x_t/u_t$ to denote the state/input of $PC_{(k,h)}$ algorithm and use $x_t^*/u_t^*$ to denote the state/input of the offline optimal. We define

$$H_t := f_t(x_t), M_t := c_t(u_{t-1}),$$
$$H_t^* := f_t(x_t^*), M_t^* := c_t(u_{t-1}^*).$$

Let $\tilde{x}_{t+k} := \tilde{\psi}_t^k(x_t^*; F)_{y_k}, \bar{x}_{t+k} = \tilde{\psi}_t^k(x_t^*, 0)_{y_k}$.

If $t \leq T - k, t \equiv 0 \pmod h$, we have

$$\|x_{t+h} - x_{t+h}^*\|^2$$
$$= \left\|\tilde{\psi}_t^k(x_t; F)_{y_h} - \psi_t^k(x_t^*, x_{t+k}^*)_{y_h}\right\|^2$$
$$\leq \left(\left\|\tilde{\psi}_t^k(x_t; F)_{y_h} - \tilde{\psi}_t^k(x_t^*; F)_{y_h}\right\| + \left\|\tilde{\psi}_t^k(x_t^*; F)_{y_h} - \psi_t^k(x_t^*, x_{t+k}^*)_{y_h}\right\|\right)^2 \tag{24a}$$
$$\leq (1+\epsilon)\left\|\tilde{\psi}_t^k(x_t; F)_{y_h} - \tilde{\psi}_t^k(x_t^*; F)_{y_h}\right\|^2 + \left(1 + \frac{1}{\epsilon}\right)\left\|\tilde{\psi}_t^k(x_t^*; F)_{y_h} - \psi_t^k(x_t^*, x_{t+k}^*)_{y_h}\right\|^2 \tag{24b}$$
$$\leq (1+\epsilon)\left\|\tilde{\psi}_t^k(x_t; F)_{y_h} - \tilde{\psi}_t^k(x_t^*; F)_{y_h}\right\|^2 + \left(1 + \frac{1}{\epsilon}\right)\left\|\psi_t^k(x_t^*, \tilde{x}_{t+k})_{y_h} - \psi_t^k(x_t^*, x_{t+k}^*)_{y_h}\right\|^2 \tag{24c}$$
$$\leq (1+\epsilon)C^2\lambda^{2h}\|x_t - x_t^*\|^2 + C^2\lambda^{2(k-1-h)} \cdot \left(1 + \frac{1}{\epsilon}\right)\|x_{t+k}^* - \tilde{x}_{t+k}\|^2 \tag{24d}$$
$$\leq \frac{1}{1+\epsilon}\|x_t - x_t^*\|^2 + C^2\lambda^{2(k-1-h)} \cdot \left(1 + \frac{1}{\epsilon}\right)\|x_{t+k}^* - \tilde{x}_{t+k}\|^2, \tag{24e}$$

where we use the triangle inequality in (24a); the AM-GM inequality in (24b); the definition of $\tilde{x}_{t+k}$ in (24c); Theorem 3.3 in (24d); the assumption of Theorem 4.3 on $h$ in (24e).

Since the objective function plus the indicator of the feasible set is $m_f$-strongly convex in variables $x_{t+1:t+k}$, we see that

$$\left\|x_{t+k}^* - \bar{x}_{t+k}\right\|^2 \leq \frac{2}{m_f} \sum_{\tau=1}^{k}(H_{t+\tau}^* + M_{t+\tau}^*). \tag{25}$$

Since $f_{t+k}$ is $m_f$-strongly convex, we also see that $\left\|x_{t+k}^*\right\|^2 \leq \frac{2}{m_f}H_{t+k}^*$. Recall that the terminal cost $F(x_{t+k}) = \alpha(\|x_{t+k}\|)$, where $\alpha$ is a class K function. By the definition of $\tilde{\psi}$, we see that $\|\tilde{x}_{t+k}\| \leq \|\bar{x}_{t+k}\|$. Therefore, we obtain that

$$\left\|x_{t+k}^* - \tilde{x}_{t+k}\right\|^2 \leq 2\|\tilde{x}_{t+k}\|^2 + 2\left\|x_{t+k}^*\right\|^2 \tag{26a}$$
$$\leq 2\|\bar{x}_{t+k}\|^2 + 2\left\|x_{t+k}^*\right\|^2 \tag{26b}$$
$$\leq 4\left\|\bar{x}_{t+k} - x_{t+k}^*\right\|^2 + 6\left\|x_{t+k}^*\right\|^2 \tag{26c}$$
$$\leq \frac{8}{m_f}\sum_{\tau=1}^{k}(H_{t+\tau}^* + M_{t+\tau}^*) + \frac{12}{m_f}H_{t+k}^* \tag{26d}$$
$$\leq \frac{20}{m_f}\sum_{\tau=1}^{k}(H_{t+\tau}^* + M_{t+\tau}^*),$$

where we used Cauchy-Schwarz inequality in (26a) and (26c); we used $\|\tilde{x}_{t+k}\| \le \|\bar{x}_{t+k}\|$ in (26b); we used (25) in (26d).

Suppose $T = n_0 \cdot h + m_0$, where $n_0 \in \mathbb{Z}_+$ and $k - h + 1 \le m_0 \le k$. Summing up the previous inequality for $t = 0, h, 2h, \ldots, (n-1)h$, we obtain that

$$\sum_{i=1}^{n_0} \left\|x_{ih} - x_{ih}^*\right\|^2 \le C^2 \lambda^{2(k-1-h)} \cdot \frac{(1+\epsilon)^2}{\epsilon^2} \cdot \sum_{i=1}^{n_0-1} \left\|x_{ih+k}^* - \tilde{x}_{ih+k}\right\|^2$$

$$\le C^2 \lambda^{2(k-1-h)} \cdot \frac{(1+\epsilon)^2}{\epsilon^2} \cdot \frac{20}{m_f} \cdot \text{cost}(OPT), \tag{27}$$

where we used (26) in the last inequality.

Therefore, we obtain that for all $\eta > 0$,

$$\text{cost}(PC_{(k,h)}) - (1+\eta)\,\text{cost}(OPT)$$

$$= \left(\sum_{i=0}^{n_0-1} \iota_{ih}^h(x_{ih}, x_{(i+1)h}) + \iota_{n_0h}^{m_0}(x_{n_0h}, x_T)\right) - (1+\eta)\left(\sum_{i=0}^{n_0-1} \iota_{ih}^h(x_{ih}^*, x_{(i+1)h}^*) + \iota_{n_0h}^{m_0}(x_{n_0h}^*, x_T^*)\right)$$

$$= \sum_{i=0}^{n_0-1} \left(\iota_{ih}^h(x_{ih}, x_{(i+1)h}) - (1+\eta)\iota_{ih}^h(x_{ih}^*, x_{(i+1)h}^*)\right) + \left(\iota_{n_0h}^{m_0}(x_{nh}, x_T) - (1+\eta)\iota_{n_0h}^{m_0}(x_{nh}^*, x_T^*)\right)$$

$$\le \sum_{i=0}^{n_0-1} \left(\iota_{ih}^h(x_{ih}, x_{(i+1)h}) - (1+\eta)\iota_{ih}^h(x_{ih}^*, x_{(i+1)h}^*)\right) + \left(\iota_{n_0h}^{m_0}(x_{nh}, x_T^*) - (1+\eta)\iota_{n_0h}^{m_0}(x_{nh}^*, x_T^*)\right) \tag{28a}$$

$$\le \left(1 + \frac{1}{\eta}\right) \cdot \frac{L+\ell_f}{2} \sum_{i=0}^{n_0-1} \left(\left\|x_{ih} - x_{ih}^*\right\|^2 + \left\|x_{(i+1)h} - x_{(i+1)h}^*\right\|^2\right) + \left(1 + \frac{1}{\eta}\right) \cdot \frac{L+\ell_f}{2} \left\|x_{n_0h} - x_{n_0h}^*\right\|^2 \tag{28b}$$

$$= \left(1 + \frac{1}{\eta}\right) \cdot (L+\ell_f) \sum_{i=1}^{n_0} \left\|x_{ih} - x_{ih}^*\right\|^2$$

$$\le \left(1 + \frac{1}{\eta}\right) \cdot (L+\ell_f) \cdot C^2 \lambda^{2(k-1-h)} \cdot \frac{20(1+\epsilon)^2}{m\epsilon^2} \cdot \text{cost}(OPT), \tag{28c}$$

where we use the fact that the PC algorithm (with replan window $h$) plans optimally after time step $nh$ in (28a); we use Corollary F.1 in (28b); we use (27) in (28c).

By setting $\eta \sim \epsilon^{-1}\left(\frac{L+\ell_f}{m}\right)^{\frac{1}{2}} \cdot C\lambda^{k-1-h}$, we see that the competitive ratio of the $PC_{(k,h)}$ algorithm (with replan window $h$) is in the order of

$$1 + O\left(\epsilon^{-1}\left(\frac{L+\ell_f}{m}\right)^{\frac{1}{2}} \cdot C\lambda^{k-1-h}\right).$$

$\square$