# OpenReview forum: "Perturbation-based Regret Analysis of Predictive Control in Linear Time Varying Systems"
_NeurIPS.cc/2021/Conference — NeurIPS 2021 Spotlight_

### Official Review · Reviewer_YZqG · 2021-06-27

**Rating:** 8
**Confidence:** 4

**Summary:**

This paper establishes both dynamic regret and competitive ratio bounds for online control with time-varying linear dynamics and adversarial perturbative noise, thereby extending past work for time-invariant noise-free dynamics. The analysis is facilitated by a general-purpose sensitive analysis for smoothed OCO.

Update: The reviewers adequately addressed my concerns, and I am ammending my review to an 8.

**Limitations And Societal Impact:**

Yes, the authors did.

**Main Review:**

Overall, I had a very positive impression of the paper. The writing and notation were consistent and clear. Authors establish guarantees both in terms of regret, and competitive ratio, the latter of which I found quite surprising. The reduction to smoothed convex optimization also seems like a useful tool. I encourage acceptance (score of 7), but am I reluctant to award a higher score because the result seems like a (somewhat) incremental extension of Li et al 2019: notably, dynamic regret with a lookahead suffers excess loss lambda^k, where lambda = sqrt(condition number). Certainly this work is more general, is considerably better written, and the analysis relies only on conditioning of the controllability Gramian (rather than Li et al. which work in the rather hideous cannonical controllable form). But the key ideas: lookahead due to banded structure (see below in Major suggestions for more explanation), and the reduction to optimization on a finite window with an extra variable x_{t+p} corresponding to a final state are both fundamentally the contributions of the prior work. In fact, I believe that the latter (the reduction from control to SOCO) could have been better attributed to the predecessor.

In sum,  I am enthusiastic about acceptance of this paper, but view it more of an improvement of past work than a fundamental breakthrough in its own right.


Major Suggestions
1. I think the sensitivity analysis based on banded diagonal matrices is super interesting. However, in light of past work (Li et al. 2019), there also seems to be an "algorithmic proof" of the same fact. Namely, I can consider the exact optimal solution to some banded optimization problem, and then I can consider the iterates produced by Nesterov's method. Examining the form of the latter, the k-th iterate requires at most k-steps of lookahead, yet exhibits an O(lambda^k) convergence to the optimal.  Then I could argue that any k-look ahead exact solution is also close to the k-step iterate of Nesterov's algorithm. Of course, "under the hood", the analysis of Nesterov's method and the conditioning of banded diagonal matrices reflect more or less the same principle. But I think that, given the use of this algorithm in past work, it would be nice to at least remark that it could be used as an alternate proof.

Minor Suggestions:
1. It took me a while to understand the subscript notation after the psi functions, only to realize that it corresponded to a vector index. I think it would be helpful to explain that upfront.
2. The authors do a good job of keeping notation consistent and avoid overloading. Still, it would have been very helpful to have seen a notation table in the appendix. This is particularly useful for disambiguating hats v.s. tildes. vs no accents, and for remembering that y_t are a state variable like x_t (which is non-standard in control, where y_t are typically system outputs).


**Time Spent Reviewing:**

2

---

> ### Author Response · Authors · 2021-08-10
> **Response to Reviewer YZqG**
>
> We want to thank the reviewer for providing the insightful comments and the constructive suggestion in ‘Major Suggestions’. We agree that Nesterov's algorithm can potentially provide an alternative algorithmic proof to our perturbation bounds in Thm 3.1 and Thm 3.3. It is interesting to see if the alternative proof can provide stronger and more general results.
>
> Although our results and techniques have some similarities with [3](Li et al., 2019), there are some key differences between them:
>
> 1. While [3] considers a linear time-invariant system, our results apply to a more general linear time-varying system.
>
> 2. Compared with the RHGC algorithms proposed by [3], the MPC-based predictive control algorithm considered in our work is more classical and requires more complicated analysis tools.
>
> 3. The reduction from online control to online optimization with memory in [3] relies on the control canonical form, which prohibits its generalization to LTV systems. In contrast, our reduction from optimal control to SOCO leverages the principle of optimality, and thus can be applied to more general dynamics.
>
> 4. Both [3] and our paper exploit the special banded structure in the time horizon, but the exploitations are done in different ways. While [3] focuses on leveraging such structure to compute the gradient of the objective locally, our perturbation bound (Thm 3.3) is about the sensitivity of the optimal solution.

---

> > ### Comment · Reviewer_YZqG · 2021-08-13
> > **Thanks for the reply!**
> >
> > Thank you for the reply. I encourage the authors to include some of their above response as a discussion in the revision. I will also be revising up my review to an 8.

---

### Official Review · Reviewer_Dpte · 2021-07-14

**Rating:** 6
**Confidence:** 3

**Summary:**

The paper considers the control of a linear time-varying system with predictions. The predictions must be accurate and describe both the transition parameters, the noise, and the cost functions. The length of the prediction window must be larger than the system's sequential strong controllability index, i.e., such that it gives the controller the ability to transition perfectly between any two given states in a fixed number of steps without exerting too much control.

The main results are that for, prediction length $k$ large enough, we get dynamic regret $O(\lambda^k T)$ and competitive ratio $1 + O(\lambda^k)$.

To achieve their results, the authors formulate the control as an optimization problem and carry out a sensitivity analysis with respect to both the start and end state as well as the perturbations. They then use this to reduce the problem to smoothed online convex optimization, which then yields the dynamic regret. They also use the sensitivity analysis together with what they describe as standard methods to achieve the competitive ratio result.

**Limitations And Societal Impact:**

Yes

**Main Review:**

Tackling the control of linear time varying systems is certainly of interest. The main algorithms and the statement of the results are satisfying and easy enough to understand. At a very high level, the proof ideas are also conveyed clearly.

While I'd be very interested to see a sensitivity analysis with respect to the prediction accuracy, it is more than reasonable to leave this for future work. I'd be happy to know if the authors have any ideas regarding this matter.

I do have a few issues:

1. While the exposition made me think the results are plausible, I did not find it sufficiently convincing. I tried looking a bit in the supplementary material, but the proofs there are quite long and technical and will take me very long to go over. Ideally, I would like to see a concrete proof sketch in the main text or perhaps even a full proof composed of informative/technical side lemmas that are proved in the appendix. In any case, it is not currently clear to me how the intermediate results in the main text come together to conclude the main results.

2. The definition of the cost seems non-standard to me. In particular, the additive relation between the state and control costs. Is it necessary? If so, while this includes the standard LQR costs, its necessity should be discussed.

3. Assumption 4 (all costs are minimized at 0 with value 0) is also non-standard. While the authors explain how it may be removed, the explanation is very short. After some thought, I find it likely that their explanation is indeed correct. Nontheless, I think it should be explained in further detail and proved formally. This would likely go to the supplementary material but I think its inclusion is necessary since the assumption is otherwise very restrictive.



Overall I think this is a very good work, but with the non-trivial caveat that I have no reasonable way to verify the veracity of the results.

**Time Spent Reviewing:**

7

---

> ### Author Response · Authors · 2021-08-10
> **Response to Reviewer Dpte**
>
> Thank you for your insightful comments. Below are our responses to your specific comments:
>
> We agree that providing a convincing intuition behind the proof is very important especially when the rigorous proof is long and complicated. We only briefly discussed the intuition behind the proof of the dynamic regret result in line 287-298. To make it more clear, we expand and reorder the original discussion below:
>
> a) Given the well-conditioned state/control costs, it suffices to bound the distance between the controller’s trajectory and the offline optimal trajectory (i.e., $\left\lVert{x_t - x_t^*}\right\rVert$) to show the dynamic regret result. See inequalities (22) and (23) in Appendix H for technical details.
>
> b) At each time step $t$, the optimal next state (under an imaginary terminal cost $F$) from the current state $x_t$ is given by $\tilde{\psi}\_t^{T-t}(x_t, w_{t:T-1}; 0)\_{y_1}$. However, reaching the optimal next state from $x_t$ requires full knowledge of the future costs, dynamics, and disturbances. Although the controller cannot reach the optimal next state due to incomplete knowledge of the future, it can leverage the predictions of future $k$ steps to decide a near-optimal control action from state $x_t$. The suboptimality, measured by the distance $ \lVert x_{t+1} - \tilde{\psi}\_t^{T-t} (x_t, w_{t:T-1}; F)\_{y_1} \rVert $, is in the order of $O(\lambda^k)$. See inequality (19) in Appendix H for technical details.
>
> c) Using the LTV perturbation bound, we can convert the per-step suboptimality bounds to a global suboptimality bound on $\left\lVert{x_t - x_t^*}\right\rVert$ that is also in the order of $O(\lambda^k)$. See inequalities (20) and (21) in Appendix H for technical details.
> In our revision, we will add these intuitions and formulate some intermediate results as lemmas to facilitate understanding.
>
> 2. The additive form of the costs are not necessary. We currently assume it for the ease of proof. Our results can be generalized to include more general costs where $x_t$ and $u_t$ are correlated. The critical assumption is that the cost function must be strongly smooth in $(x_t, u_t)$ and strongly convex in $x_t$, as in Lemma C.1, so that the optimal control problem in the LTV system can be reduced to SOCO.  We will add a discussion of this in our revision.
>
> 3. We will add a more detailed explanation about why Assumption 4 can be made without loss of generality in revision. The key idea behind Assumption 4 is that, when the minimizer of the cost function for the next step is known, we can perform a translation in the state or control space to overlap the origin with the minimizer. We will also point the reader to the trajectory tracking example (Example 2.1) which applies the reduction to recenter the minimizers of costs to zero.

---

### Official Review · Reviewer_BGJX · 2021-07-16

**Rating:** 8
**Confidence:** 3

**Summary:**

This paper examines the control of linear time varying systems in which the controller is given predictions of the system matrices, cost functions, and disturbances for a horizon in the future. This setup is motivated by problems, such as power systems, in which reasonably accurate predictions of future dynamics and disturbances are available at least over short horizons. The paper gives two algorithms, one based on model predictive control (MPC), which replans at each time step, and another that only replans once over fixed windows. They bound the dynamic regret and competitive ratios of these algorithms, respectively. For the MPC method, the key technique bounding how the solutions to the optimization sub-problems change when the initial condition, the disturbances, and any possible terminal constraints vary. They use these bounds to analyze stability and dynamic regret in a unified manner. For competitive ratio bounds, they show how the particular algorithm can be viewed as solving a smoothed online convex optimization problem. Then they use the previously derived methods to bound the deviation of the solution they obtained from the optimal solution.

**Limitations And Societal Impact:**

The authors address the limitations of their method adequately. I see no potential negative societal impact.

**Main Review:**

This paper describes two methods for controlling linear time-varying systems when finite-horizon predictions of the dynamics, costs, and disturbances are known. While these assumptions limit the scope, the authors convincingly argue that they are realistic in some important scenarios. The paper is insightful and the proofs appear to be rigorous. The key contributions enable precise bounds on stability and suboptimality for these time varying systems when only limited preview information is available. I consider these contributions to be substantial.

The methodology relies on heavily on smoothness and strong convexity of the losses and linearity of the constraints. The assumptions, in particular, guarantee that solutions to the optimal control sub-problems become smooth functions of the problem data. As pointed out by the authors, this precludes the things like inequality constraints on the states or inputs, as the smoothness properties would be lost. In many scenarios, however, these assumptions seem reasonable.

As a minor comment, some of the notation is confusing. For example, sometimes the optimal trajectories are indexed by $h$, and other times by $y_h$. The distinction is never explained, but from reading the proofs, they appear to mean the same thing.

**Time Spent Reviewing:**

2

---

> ### Author Response · Authors · 2021-08-10
> **Response to Reviewer BGJX**
>
> Thank you for your insightful comments. We are sorry for the confusion caused by the notation. We use the subscript $h$ for the decision points in SOCO (Section 3.1), and use the subscripts $y_h$ and $v_h$ for the states and control actions in LTV (Section 3.2). We decided to use different subscripts for SOCO decision points and LTV states in order to distinguish them better in the reduction from LTV to SOCO.  In our revision, we will clarify this difference, and we will also add a table of notations to the appendix to facilitate understanding. We also plan to use another letter instead of $\hat{\psi}$ to represent the optimal SOCO trajectory to distinguish it better from the optimal LTV trajectory.

---

> > ### Comment · Reviewer_BGJX · 2021-08-25
> > **Clearing up notation would be great**
> >
> > I think a table of notation would b very helpful, as would be using distinct letters to avoid overloading. Thanks!

---

### Official Review · Reviewer_8aU4 · 2021-07-19

**Rating:** 6
**Confidence:** 2

**Summary:**

This article studies finite-time horizon   predictive control for linear systems with time-varying and strongly convex costs.
The authors assume that each time points,
 the system coefficients and
the random disturbance of the state dynamics
at the next $k$ time steps can be observed exactly,
based on which the controller choose the current action.
By reformulating the control problem as an equivalent optimisation problem,
the authors establish a perturbation bound of the algorithm,
and further analyze the dynamic regret and competitive ratio.


**Ethics Review Area:**

["I don’t know"]

**Limitations And Societal Impact:**

Yes.

**Main Review:**


My main comments are as follows:

\begin{enumerate}

\item The analysis based on the assumption that
at each time point,
 the system coefficients and  random disturbances in the future $k$ time steps
can be observed exactly for a sufficiently large $k$, depending implicitly on the controllability index of the system.
In my opinion, this is a very strong assumption, especially the exact observation of noise.
I suggest the authors to
discuss extensions to inaccurate observations of noises,
or
give more concrete and practical examples where the random noises in the future time steps
can be exactly observed.

\item
In the optimization problem (2), the authors introduce the terminal cost $F(x)=\alpha(\|x\|)$ with $\alpha$ being a convex function. However, the results and assumptions of Theorem 3.3 seem to be independent of $F$. Please clarify the role of $F$. Can we choose $F$ to be the zero function in the algorithm?

\item I recommend to recall the definition of competitive ratio in the main text for the reader's convenience.

\item The authors mention in the introduction that the current framework can not handle control constraints. However, it seems that the essential step of the analysis is to establish the Lipschitz stability of the optimizer of a convex optimization problem with respect to the loss function. Is it possible to follow the framework in https://arxiv.org/pdf/2104.09311.pdf, and handle the control constraints as an additional nonsmooth convex cost?
 \end{enumerate}


**Time Spent Reviewing:**

1.5 hours

---

> ### Author Response · Authors · 2021-08-10
> **Response to Reviewer 8aU4**
>
> Thank you for your insightful comments. Below are our responses to your specific comments:
>
> 1. As we discussed in line 95-96, assuming the predictions of future disturbances are exact is a gentle start towards understanding the more general case with prediction errors, and it is common in the literature [4, 24-26]. We also want to emphasize that the disturbance sequence in our setting is not purely random for many applications. For instance, when converting the trajectory tracking problem (Example 2.1) to our setting, the new disturbance sequence $\tilde{w_t}$ is largely decided by the trajectory to track (line 142), which makes near-exact predictions possible. Further, to generalize our analysis to include the inexact predictions, the LTV perturbation bound (Thm 3.3) already implies that the prediction errors in the far future only has a very small impact on the current control decision. However, whether the exponentially decay regret bound still holds will rely heavily on what assumptions we make on the prediction errors. So we leave this as future work.
>
> 2. The zero function satisfies our assumptions on the terminal cost $F$ (line 167). Thus, both the dynamic regret result (Thm 4.2) and the competitive ratio result (Thm 4.3) holds when $F$ is the zero function. The terminal cost $F$ is commonly used to regulate the terminal states in the literature of Model Predictive Control. It has a similar role as function $g$ defined in Section 2.1 of [Guo et al., 2021]. Many previous works on MPC (e.g. [4, 7]) need to make certain assumptions on $F$ to provide theoretical guarantees.
>
> 3. In our revision, we will recall the definition of competitive ratio in both Section 2 and Section 4.2.
>
> 4. [Guo et al., 2021] considers a different setting where the time is continuous and the disturbances are stochastic, and it only proves Lipschitz stability (Thm 2.5) rather than the required exponential decay property, which is relatively trivial in our setting. The complexity of their analysis is mainly attributed to the jump-diffusion noises, rather than the LTV property of the system. We find the technique used to deal with non-smooth costs in [Guo et al., 2021] can potentially be used to handle constraints in our setting by viewing the constraints as indicator cost functions. We will add this discussion in revision. Moreover, the coupled-system approach provides a new insightful perspective, and the analysis framework for the influence of prediction errors is also helpful to further studies.
>
> [Guo et al., 2021] Guo, Xin, Anran Hu, and Yufei Zhang. "Reinforcement learning for linear-convex models with jumps via stability analysis of feedback controls." arXiv preprint arXiv:2104.09311 (2021).

---

### Decision · Program_Chairs · 2021-09-27

**Decision:**

Accept (Spotlight)

**Comment:**

The reviewers were unanimous in their appreciation of the paper and hence I recommend a clear Accept. I request the authors to look into improving the notation in the paper and re-assessing the paper in terms of clarity of presentation of the proofs etc to improve readability. Suggestions of this form have been laid out in multiple reviews.